# TGS1 mediates 2,2,7-trimethyl guanosine capping of the human telomerase RNA to direct telomerase dependent telomere maintenance

Valentina Buemi[1,10,11], Odessa Schillaci[1,11], Mariangela Santorsola [1], Deborah Bonazza[2],
Pamela Veneziano Broccia[1], Annie Zappone[1], Cristina Bottin[3], Giulia Dell'Omo[4], Sylvie Kengne[1],
Stefano Cacchione [5], Grazia Daniela Raffa [5], Silvano Piazza[6], Fabrizio d'Adda di Fagagna [4,7],
Roberta Benetti [8], Maurizio Cortale[9], Fabrizio Zanconati[2,3], Giannino Del Sal [1,4,6] & Stefan Schoeftner [1✉]

Pathways that direct the selection of the telomerase-dependent or recombination-based, alternative lengthening of telomere (ALT) maintenance pathway in cancer cells are poorly understood. Using human lung cancer cells and tumor organoids we show that formation of the 2,2,7-trimethylguanosine (TMG) cap structure at the human telomerase RNA 5′ end by the Trimethylguanosine Synthase 1 (TGS1) is central for recruiting telomerase to telomeres and engaging Cajal bodies in telomere maintenance. TGS1 depletion or inhibition by the natural nucleoside sinefungin impairs telomerase recruitment to telomeres leading to Exonuclease 1 mediated generation of telomere 3′ end protrusions that engage in RAD51-dependent, homology directed recombination and the activation of key features of the ALT pathway. This indicates a critical role for 2,2,7-TMG capping of the RNA component of human telomerase (hTR) in enforcing telomerase-dependent telomere maintenance to restrict the formation of telomeric substrates conductive to ALT. Our work introduces a targetable pathway of telomere maintenance that holds relevance for telomere-related diseases such as cancer and aging.

[1] Dipartimento di Scienze della Vita, Università degli Studi di Trieste, Via E. Weiss 2, 34127 Trieste, Italy. [2] Struttura Complessa di Anatomia ed Istologia Patologica, Azienda Sanitaria Universitaria Giuliano Isontina (ASUGI) Trieste, Strada di Fiume 447, 34149 Trieste, Italy. [3] Dipartimento Universitario Clinico di Scienze Mediche Chirurgiche e della Salute, Università degli Studi di Trieste, Ospedale di Cattinara - Strada di Fiume 447, 34149 Trieste, Italy. [4] IFOM Foundation-FIRC Institute of Molecular Oncology Foundation, Milan 20139, Italy. [5] Dipartimento di Biologia e Biotecnologie, Sapienza Università di Roma, Roma, Italy. [6] International Centre for Genetic Engineering and Biotechnology (ICGEB), Area Science Park - Padriciano, 34149 Trieste, Italy. [7] Istituto di Genetica Molecolare, Consiglio Nazionale delle Ricerche (IGM-CNR), Pavia 27100, Italy. [8] Dipartimento di Area Medica (Dame), Università degli Studi di Udine, p.le Kolbe 4, 33100 Udine, Italy. [9] Struttura Complessa di Chirurgia Toracica, Azienda Sanitaria Universitaria Giuliano Isontina (ASUGI) Trieste, Strada di Fiume 447, 34149 Trieste, Italy. [10]Present address: Cancer Research UK Cancer Therapeutics Unit, The Institute of Cancer Research, London SM2 5NG, United Kingdom. [11]These authors contributed equally: Valentina Buemi, Odessa Schillaci. ✉email: sschoeftner@units.it

The reactivation of telomere-maintenance mechanisms (TMMs) represents a hallmark of human cancer and is critical to overcome replicative senescence, triggered by critically short telomeres[1,2]. Approximately 85% of human cancers reactivate telomerase to replenish telomere repeats at chromosome ends[2]. Telomerase-negative cancers use the alternative lengthening of telomeres pathway (ALT) to ensure telomere maintenance in a homologous recombination-dependent manner[3,4]. Both TMMs show specific association to distinct subnuclear compartments. Cajal bodies (CBs) are critical for telomerase-complex maturation and telomerase-dependent TMM[5]. Telomeres localize to CBs when telomerase is catalytically active and mutations that impair localization of telomeres to CBs are linked with defects in the recruitment of telomerase to telomeres, accelerate telomere shortening, and promote premature aging syndromes[6–9]. In contrast, telomerase-independent telomere maintenance is triggered by replication stress at telomeres that promotes the formation of recombinogenic telomere substrates that engage in intra- and intertelomeric recombination and break-induced replication events (BIR), that are concentrated to ALT-associated PML bodies (APBs)[3,10–12]. Both TMMs can coexist in a mosaic pattern in tumors but also in individual tumor cells[13–16]. Further, genetic or pharmacological disruption of telomerase activity provokes a switch to ALT in cancer cells[17–20]. This suggests the existence of biological pathways that may direct the usage of TMMs in cancer cells[17–20]. Here, we show that 2,2,7-trimethylguanosine (TMG) capping of the human telomerase RNA by the CB protein trimethylguanosine synthase 1 (TGS1) drives telomerase-based TMM by suppressing exonuclease-1 (EXO1)-dependent formation of recombinogenic telomere substrates that fuel the ALT pathway.

## Results

### Sinefungin inhibits TGS1-mediated 2,2,7-TMG capping of hTR.
Genetic depletion of TGS1 in *S. cerevisiae* results reduced 2,2,7-TMG capping of the yeast telomerase RNA component TLC1[21]. However, the role of human TGS1 in controlling telomere-maintenance pathways and telomere-strand homeostasis in cancer cells is not known. We therefore eliminated TGS1 function in H1299 lung cancer cells and lung adenocarcinoma-derived tumor organoids by enzymatic inhibition or transient siRNA-mediated knockdown. TGS1 knockdown in H1299 cells results in a 5-fold reduction of 2,2,7-TMG capping of the human telomerase RNA (hTR), as determined by RNA immunoprecipitation (RIP) using an anti-2,2,7-TMG-specific antibody (Fig. 1a–c). Anti-TMG RIP resulted in a 670-fold enrichment of hTR, but lacked enrichment of 7-monomethylguanosine-capped mRNAs such as TRF2 or actin, validating antibody specificity (Supplementary Fig. 1a–c). Inhibition of TGS1 by sinefungin, a natural nucleoside and structural analog of S-adenosyl methionine (SAM), was reported to reduce 2,2,7-TMG methylation of different viral RNAs[22,23]. We found that a 10-day treatment of H1299 cells with sinefungin resulted in a progressive decrease of TGS1 protein levels, without altering mRNA levels (Fig. 1d, e). Reduced TGS1 protein levels were rescued upon withdrawal of specific siRNA against TGS1 or sinefungin from the medium (Supplementary Fig. 4a, b). Targeting TGS1 did not alter the expression of control genes such as TRF2 or TRF1 (Supplementary Fig. 1d–k). The selective reduction of TGS1 protein but not mRNA expression triggered by sinefungin is reminiscent to a phenomenon referred to as "targeted protein degradation", as previously described for selective estrogen-receptor downregulators (SERDs) that trigger proteosomal degradation of the estrogen receptor[24,25]. In line with reduced TGS1 protein levels, sinefungin treatment of H1299 cells resulted in a significant, 4-fold reduction of 2,2,7-TMG capping of hTR (Fig. 1f). Sinefungin treatment did not impact on the

capping of mRNAs such as TRF2 and actin (Supplementary. Fig. 1l–n). This identifies sinefungin as an efficient TGS1 inhibitor that interferes with hTR maturation by blocking 2,2,7-TMG cap formation.

To validate the effect of sinefungin in an independent lung cancer model, we generated tumor organoids from surgically resected human lung adenocarcinomas. Immunohistochemistry analyses and H&E staining revealed conserved marker expression and morphology between parental tumor tissue and derived organoids (Fig. 1g–i). Whole-exome sequencing analysis in three matched primary lung tumors, adjacent nontumor tissues, and cultivated tumor organoids revealed that tumor organoids retained mutations in cancer-relevant genes, such as PIK3CA, PTEN, and CDKN1A, originally found in parental tumor tissues (Fig. 1j, see "Methods" for details). This confirms the tumor origin of used organoids. Sinefungin-treated and -untreated tumor organoids displayed normal growth during the chosen experimental time window (Fig. 1k–m). Finally, sinefungin treatment resulted in a reduction of TGS1 protein but not mRNA, recapitulating experimental data from H1299 cells (Fig. 1n–q).

Together, this demonstrates that the S-adenosylmethionine analog sinefungin is an efficient antagonist of TGS1-dependent 2,2,7-TMG capping of hTR in monoclonal cancer cell lines but also in patient-derived 3D cell culture model systems.

### TGS1 stabilizes telomerase-dependent telomere maintenance.
Cajal bodies (CBs) play a central role in telomerase-complex maturation, and transient localization of telomeres to CBs was reported to promote telomerase-dependent telomere elongation[6]. Given that TMG capping of hTR by TGS1 is thought to occur in CBs, we investigated features of telomerase-dependent telomere maintenance in experimental cells[26]. Sinefungin treatment of H1299 cells and lung tumor organoids significantly reduced CB number (Supplementary Fig. 2b, c). These data are supported by TGS1- knockdown experiments using H1299 cells (Supplementary Fig. 2a). We found that the association of telomeres to the remaining CBs was significantly reduced, both in sinefungin-treated or TGS1- knockdown model H1299 cells and tumor organoids (Fig. 2a–h). This indicates an impairment of CB-based pathway of telomere maintenance in TGS1 loss-of-function cells. Applying the telomere-repeat amplification protocol (TRAP), we found significantly increased in vitro telomerase activity in cell lysates from H1299 cells after transient, RNAi-mediated depletion of TGS1 or sinefungin treatment (Supplementary Fig. 3a, b). This recapitulates data from TGS1-knockdown HeLa cells (ref. [27]) and demonstrates that 2,2,7-TMG capping is not essential for the enzymatic activity of the telomerase complex in vitro. Northern blotting and qRT-PCR on TGS1-knockdown H1299 cells and sinefungin-treated H1299 cells or patient-derived organoids displayed a small but consistent increase of hTR expression without affecting the expression of control genes such as hTERT, TRF2, or TRF1 on the protein or mRNA level (Supplementary Fig. 3c–i; Supplementary Fig. 1d–k). This finding supports a previous study that links increased hTR expression and elevated in vitro TRAP activity in HeLa cells upon genetic depletion of TGS1[27]. Loss of the Cajal body component TCAB1 was demonstrated to impair telomerase recruitment to telomeres without altering overall telomerase activity[28–30]. We were therefore interested in investigating as to whether loss of hTR 2,2,7-TMG capping in CBs impairs the localization of telomerase to telomeres and to CBs. "Super-telomerase" HeLa or HT1080 cells that ectopically overexpress both, hTR and hTERT, represent an accepted model system to study the functional relationship between telomeres, telomerase, and Cajal bodies[6,29]. We generated H1299 cell clones

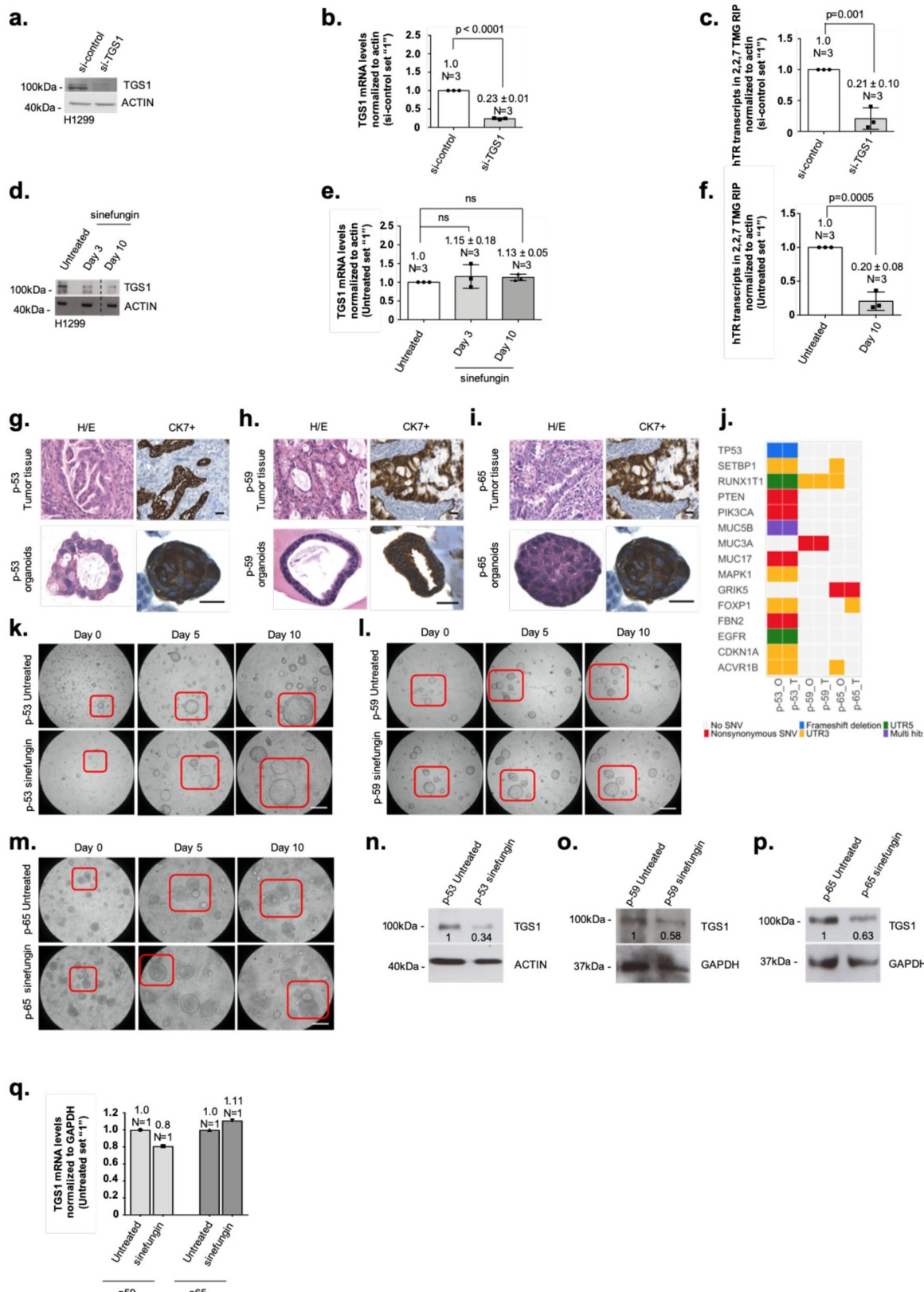

stably overexpressing a FLAG-epitope-tagged version of the catalytic subunit of telomerase, hTERT (clone-9 cells) and a derivate — H1299 clone ectopically overexpressing FLAG-hTERT and hTR (clone-22 cells) (Supplementary Fig. 2d, e). FLAG-hTERT localizes to clone-22 telomeres as shown by chromatin immunoprecipitation (ChIP) and immunofluorescence (Supplementary Fig. 2f, g). This is paralleled by increased overall telomere length and augmented colocalization of telomeres with CBs (Supplementary Fig. 2h–j). Transient knockdown of TGS1 in clone-22 cells was found to significantly decrease the

**Fig. 1 Inhibition of TGS1 blocks 2,2,7-TMG capping in preclinical lung cancer models. a** Western blotting of proteins from H1299 cells, transfected with the indicated siRNAs for 10 days. Immunoblots were probed with indicated antibodies. **b** Quantification of TGS1 mRNA levels by qRT-PCR in H1299 cells, transfected with indicated siRNAs for 10 days. Values were normalized against actin. **c** RNA-immunoprecipitation experiments using anti-2,2,7-TMG antibody followed by qRT-PCR. Total RNA was prepared from H1299 cells transfected with indicated siRNAs. **d** Western blotting of proteins from H1299 cells treated as indicated. Immunoblots were probed with indicated antibodies. **e** Quantification of TGS1 mRNA by qRT-PCR in H1299 cells treated as indicated. Values were normalized against actin. **f** RNA-immunoprecipitation experiments using anti-2,2,7-TMG antibody followed by qRT-PCR. Total RNA was prepared from H1299 cells treated, as indicated. **g–i** Immunohistochemistry on FFPE sections of lung tumor organoids and corresponding primary tumors. Hematoxylin and eosin (H/E) staining (violet) and CK7 immunostaining (brown) are shown. Scale bar, 5 μm. **j** Oncoprint of selected somatic tumor-specific SNVs and indels affecting cancer-relevant genes and shared between tumor tissues and organoid pairs. Variations include nonsynonymous mutations (e.g., PIK3CA, PTEN), UTR-site mutations (e.g., CDKN1A, MAPK1), and frameshift indels (e.g., TP53). Rows and columns represent biological samples (O, tumor organoid; T, tumor tissue) and genes, respectively. Type of sequence alterations is indicated by a color code. **k–m** Representative pictures of organoids generated from lung tumor tissue treated as described. Red rectangle indicates growth of organoids cultured over time. Scale bar, 10 μm. **n–p** Western blotting of proteins from lung tumor organoids treated as described for 10 days. Values of densitometric analysis are shown. Untreated samples were set to "1". **q** Quantification of TGS1 mRNA levels by qRT-PCR in two lung tumor organoids treated as described for 10 days. Values were normalized against GAPDH. Bars in panels **b**, **c**, **e**, **f** indicate mean values. N = number of independent experiments; whiskers indicate standard deviation. A two-tailed Student's t-test was used to calculate p-values. Uncropped blots in Source Data.

colocalization frequency of FLAG-hTERT with telomeres, as demonstrated by co-immunofluorescence analysis using anti-FLAG and anti-TRF1 antibodies (Fig. 2i, j). In line with this, hTR RNA-FISH combined with anti-TRF2 immunofluorescence revealed a significantly reduced recruitment of the telomerase RNA component to chromosome ends in TGS1-knockdown cells (Fig. 2k, l). Impaired recruitment of FLAG-hTERT to telomeres was confirmed by anti-FLAG telomere ChIP in sinefungin-treated clone-22 cells (Fig. 2m, n). Together, our data indicate that TGS1-dependent maturation of hTR is not essential for in vitro telomerase activity, but represents an important factor in stabilizing telomere–CB interactions and efficient recruitment of telomerase to telomeres.

**TGS1 regulates telomere single-strand homeostasis in cancer cells.** To investigate the impact of TGS1 loss of function on telomere-length homeostasis, we performed quantitative DNA-FISH (Q-FISH) on G-rich and C-rich telomere strands in H1299 cells and lung tumor organoids. Quantitative DNA-FISH analysis using telomere-strand-specific probes revealed a 10 and 15% increase of the G-rich telomere-strand length after 10 days of TGS1 knockdown or sinefungin treatment (Fig. 3a, b). Of note, reconstituting TGS1 function, by suspending sinefungin treatment or replacing TGS1-specific siRNAs with control siRNAs for an additional 10 days rescued telomere G-rich strand length to normal levels (Fig. 3a, b; Supplementary Fig. 4a, b). In contrast with data from the G-rich telomere strand, we found that siRNA-mediated knockdown of TGS1 or sinefungin treatment of H1299 cells for 10 days led to a 38% or 42% decrease of C-rich telomere-strand length, respectively (Fig. 3c, d). Again, alterations in C-rich telomere-strand length were rescued to normal levels after reconstitution of TGS1 function (Fig. 3c, d). Elongated G-rich telomere-stand overhangs in H1299 TGS1 loss-of-function cells were validated by native telomere-restriction fragment analysis (Supplementary Fig. 4c, d). Sinefungin treatment of lung tumor organoids recapitulated telomere G-rich strand elongation and efficient C-rich telomere-strand shortening in an independent preclinical model system for lung cancer (Fig. 3e–i). Our data suggest that lack of telomerase recruitment to telomeres in TGS1 loss-of-function cells leads to downstream events that alter the length of telomere single strands. To find additional evidence for such a model, we mimicked a lack of telomerase at telomeres by acute, RNAi-mediated depletion of hTERT in our H1299 cell model systems. We found that transient hTERT knockdown in H1299 but also clone-22 cells recapitulated alterations in the length of G-rich

and C-rich telomere strands observed in sinefungin-treated H1299 and TERT-depleted clone-22 cells (Supplementary Fig. 4e–l). As expected, siRNA-mediated depletion of hTERT in sinefungin-treated H1299 or TGS1-depleted clone- 22 cells did not exacerbate telomere-strand-length alterations (Supplementary Fig. 4e–l).

Altogether, our data demonstrate that loss of TGS1 function results in the formation of extended G-rich telomere single-strand protrusions in standard and advanced preclinical lung cancer model systems. This identifies TGS1-dependent 2,2,7-TMG of hTR as a novel regulator of telomere single-strand homeostasis in human cancer cells.

**Loss of TGS1 promotes APB formation.** In vertebrates, multiple exonucleases were demonstrated to have a role in controlling the length of G-rich or C-rich telomere single strands to ensure vertebrate telomere function, maintenance, and replication[31,32]. An extended length of G-rich or C-rich telomere-strand overhang length has been linked with strand invasion and homology-directed recombination in ALT cells[33,34]. We therefore tested whether TGS1-mediated 2,2,7-TMG capping of hTR promotes telomerase-dependent TMM by restricting the formation of recombinogenic G-rich telomere-strand substrates that engage in ALT in telomerase- positive lung cancer cells. We found that both TGS1 knockdown and sinefungin treatment resulted in elevated PML body numbers and increased colocalization frequency of PML bodies with telomeres in both H1299 cells and lung tumor organoids. Together, this suggests that loss of TGS1 function promotes the formation of APBs, a key feature of ALT cells (Fig. 4a–h; Supplementary Fig. 5a–c). In line with data from telomere single-strand analysis, loss of hTERT promotes APB formation, but does not exacerbate sinefungin or siTGS1-mediated increase in telomere–PML-body colocalization frequency (Supplementary Fig. 5d–f). Increased APB levels in TGS1 loss-of-function cells were paralleled by a significant increase in telomere sister chromatid-exchange (T-SCE) rates, as determined by sister chromatid orientation FISH using differentially labeled, telomere-strand-specific probes. In this experimental setup, siRNA-mediated depletion of RAD51 rescued increased T-SCE frequency in TGS1-knockdown H1299 cells to control levels (Fig. 4i, j). Our data demonstrate that loss of TGS1 function leads to extended G-rich telomere- strand overhangs that engage in RAD51-dependent homologous recombination events that are paralleled by a shift of telomere localization from CB to PML bodies, indicative for an ectopic activation of the ALT pathway.

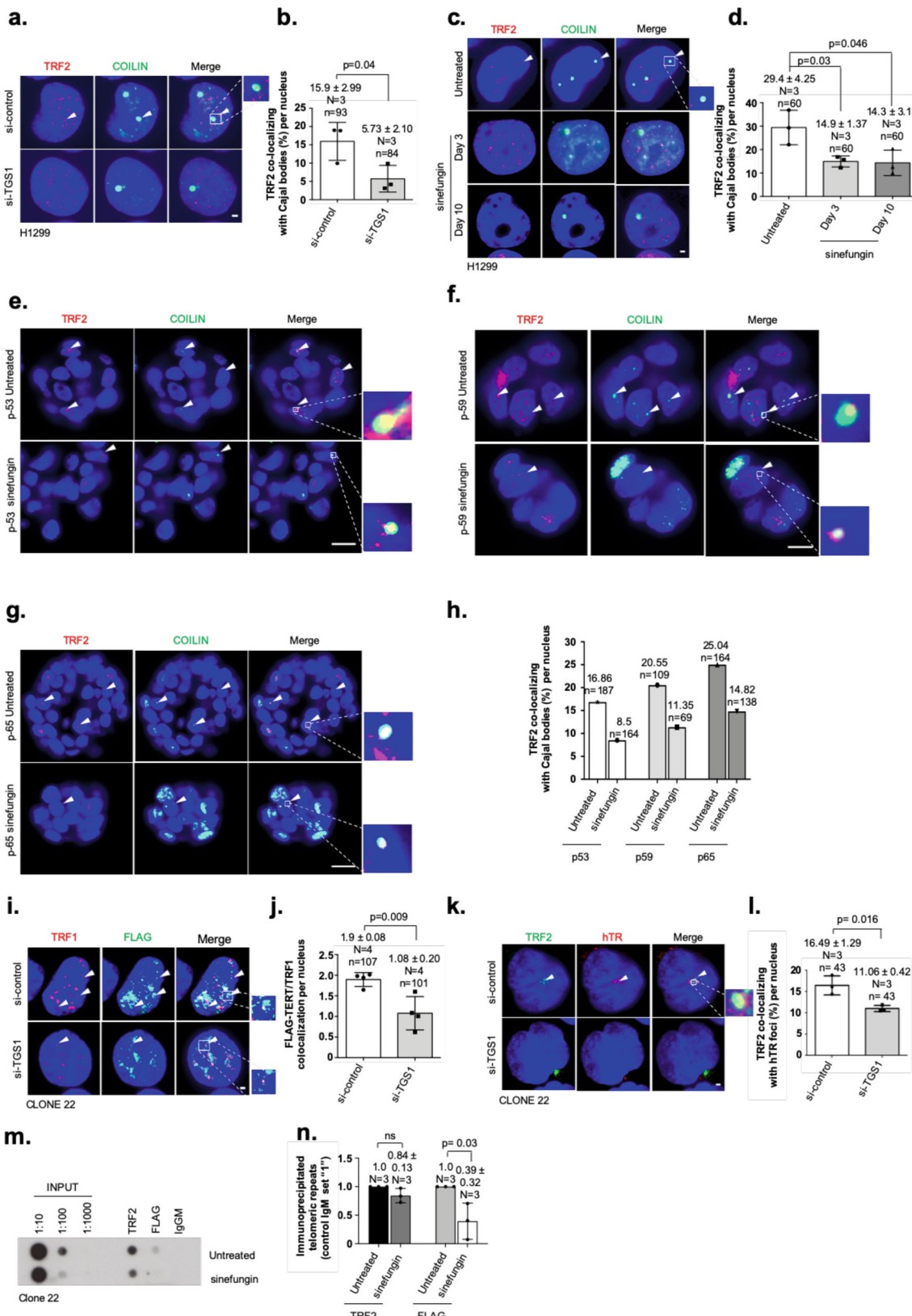

**TGS1 suppresses key features of ALT in telomerase-positive cancer cells**. We next aimed to find additional evidence that supports a critical role for 2,2,7-TMG capping of hTR in antagonizing the activation of the ALT pathway in telomerase-positive

cells. Telomere replication stress, triggered by DNA:RNA hybrids, was demonstrated to promote the formation of recombinogenic telomere substrates that engage in homologous recombination in ALT cell lines[35]. siRNA-mediated depletion of TGS1 leads to

**Fig. 2 TGS1 directs telomerase-dependent telomere maintenance involving Cajal bodies. a** Representative images of combined immunofluorescence with anti-TRF2 and anti-Coilin antibodies on H1299 cells transfected with indicated siRNAs for 10 days. **b** Percentage of Cajal bodies colocalizing with telomeres (TRF2) of the experiment shown in **a**. **c** Representative images of combined immunofluorescence with anti-TRF2 and anti-Coilin antibodies on H1299 cells treated, as described. **d** Percentage of Cajal bodies colocalizing with telomeres (TRF2) of the experiment shown in **c**. **e–g** Representative images of combined immunofluorescence with anti-TRF2 and anti-Coilin antibodies in tumor lung organoids treated as described. **h** Percentage of Cajal bodies colocalizing with telomeres (TRF2) on lung tumor organoids shown in e–g. **i** Representative images of combined immunofluorescence with anti-TRF1 and anti-FLAG antibodies from clone-22 cells transfected with indicated siRNAs. **j** Quantification of TRF1/FLAG-hTERT colocalization events of the experiment shown in **i**. **k** Representative images of hTR RNA-FISH combined with immunofluorescence using anti-TRF2 antibody on H1299 cells transfected with indicated siRNAs for 10 days. **l** Percentage of hTR colocalizing with telomeres (TRF2) per nucleus of cells shown in **k**. **m** Chromatin immunoprecipitation experiments (ChIP) using clone-22 cells treated, as described for 10 days. Mouse anti-TRF2 and mouse anti-FLAG antibodies were used. Mouse control IgM was used as negative control. Serial dilutions of chromatin extract (input) prepared from clone-22 cells, treated as described, were loaded. **n** Quantification of three independent ChIP experiments, average enrichment of telomeric repeats is indicated. For quantifications in **b**, **d**, **h**, **j**, **l**, and **n**, mean values are indicated; whiskers indicate standard deviation; N = number of independent experiments. n = number of analyzed nuclei. Arrowheads indicate colocalization events. Bars in panels **b**, **d**, **j**, **l**, and **n** indicate mean values. Red and green, epitopes detected by immunofluorescence; blue, DAPI-stained nuclei. Scale bars in **a**, **c**, **i**, **k** correspond to 1 µm. Scale bars in **e–g** correspond to 10 µm. A two-tailed Student's *t*-test was used to calculate statistical significance; *p*-values are shown. Uncropped blots in Source Data.

increased DNA:RNA hybrid levels at telomeres, as detected by immunofluorescence using a specific S9.6 monoclonal antibody (Fig. 5a, b). This result was reproduced when performing telomere DNA:RNA hybrid immunoprecipitation (DRIP) analysis performed using a radiolabeled telomere-specific probe (Fig. 5c). Replication stress is linked with the activation of ATR that promotes C-circle formation and homologous recombination at telomeres[36–39]. In line with this, knockdown of TGS1 in H1299 cells leads to increased localization of phosphorylated ATR (Ser 428) at telomeres that is paralleled by RAD51 recruitment to telomeres (Fig. 5d–g, Supplementary Fig. 6a, b). Loss of TGS1 also increased the localization of RAD51 to PML bodies (Fig. 5h, i). This is indicative for elevated ALT activity and explains the rescue of telomere sister chromatid exchange to control levels in TGS1– RAD51 double-knockdown cells (Fig. 4i, j). Bloom helicase (BLM) is an integral component of the ALT pathway, that was shown to locate to ALT telomeres, initiate telomere clustering, and promote the association of extrachromosomal telomere repeats (ECTRs) with chromosome ends[40–42]. RNAi-mediated depletion of TGS1 results in a significant increase in colocalization frequency between BLM foci and the telomere marker TRF1, consistent with elevated ALT activity (Fig. 5j, k, Supplementary Fig. 6c). Genetic mutations and epigenetic alterations at telomeres of ALT cells determine the presence of ECTRs, typically present as circular DNA molecules characterized by a continuous C-rich strand with nicks or deletions in the complementary strand[43]. C-circles are readily detectable in U-2 OS cells, a classic ALT cell line, and are almost undetectable in telomerase-positive H1299 cells (Fig. 5l). However, we found that RNAi-mediated depletion of TGS1 mediated a significant increase in C-circle formation in experimental cells (Fig. 5l, m). Together, our data indicate that TGS1 has a role in antagonizing the activation of the ALT pathway in telomerase-positive cancer cells.

**TGS1 limits ALT by protecting from EXO1-mediated telomere resection.** Pronounced shortening of the C-rich telomere strand in TGS1 loss-of-function models suggests the recruitment of exonuclease activity to telomeres. Recently, exonuclease 1 (EXO1) has been reported to act on telomeres, generating a 3′ overhang[44]. In *S. cerevisiae*, EXO1 was shown to mediate telomere resection and homologous recombination in the absence of telomerase[45–47].

We therefore performed strand-specific telomere-length measurements in H1299 cells depleted for TGS1 or both TGS1 and EXO1. As expected, loss of TGS1 led to a significant increase of the G-rich telomere strand and decreased length of the C-rich

telomere strand (Fig. 6a–d). Single-strand telomere Q-FISH analyses further revealed that loss of EXO1 rescues G-rich telomere and C-rich telomere-strand alterations in TGS1-knockdown H1299 cells (Fig. 6a–d). In line with these results, we found that knockdown of EXO1 in TGS1 loss-of-function cells reconstituted normal CB-body numbers but also rescued colocalization frequency between CBs and telomeres to control levels (Fig. 6e, f; Supplementary Fig. 7a). Accordingly, loss of EXO1 abolished the increased localization of telomeres to PML bodies in H1299 cells lacking TGS1 (Fig. 6g, h; Supplementary Fig. 7b). Together, this indicates that telomere C-rich strand resection by EXO1 in TGS1 loss-of-function cells is important to direct a switch in the usage of telomere-maintenance compartments, shifting a fraction of telomeres from CB to PML bodies.

Our data show that EXO1-dependent generation of 3′ telomere-strand overhangs is linked with the emergence of ALT features that were, however, not sufficient to result in a length gain of both telomere strands (Fig. 6a–d). We thus hypothesized that the APB compartment in TGS1 loss-of-function cells is not sufficiently strong to support telomere elongation by ALT. Short-term treatment of cancer cells with the inhibitor of DNA methylation, 5-Aza-2′-deoxycytidine, was shown to increase T-SCE frequency and APB numbers in cancer cell lines, including H1299 cells[48,49]. We therefore wished to test whether stimulation of the ALT pathway by combined use of sinefungin and 5-Aza-2′-deoxycytidine converts telomere single-strand alterations of TGS1 loss-of-function cells into an overall increase in double- stranded telomere repeats. We found that increased APB numbers of sinefungin-treated H1299 cells further augmented upon the addition of 5-Aza-2′-deoxycytidine (Fig. 6i, j; Supplementary Fig. 7c). This effect was paralleled by an enhanced recruitment of the BLM helicase to telomeres and C-circle formation (Supplementary Fig. 7d–h). Strand-specific telomere-length measurements revealed that 5-Aza-2′-deoxycytidine treatment was able to convert the shortening of the C-rich telomere strand in sinefungin- treated cells into a 53% telomere-length increase in a combined treatment scheme, when compared with control cells (Fig. 6m, n). This effect was paralleled by an efficient, 40% increase in G-rich telomere- strand length (Fig. 6k, l). Separation of telomeres into 4 quartiles according to fluorescence intensity (arbitrary fluorescence units, a.f.u) revealed an enrichment of C-rich and G-rich telomere strands in the quartile representing the category of longest telomeres (top 25%, Q4) and reduction of telomeres representing the category of shortest telomeres (bottom 25%, Q1, Supplementary Fig. 7i, j). RAD51 and BLM were recently demonstrated to promote the recruitment of the DNA-polymerase δ-subunit POLD3 to telomeres in ALT cells[50]. In this

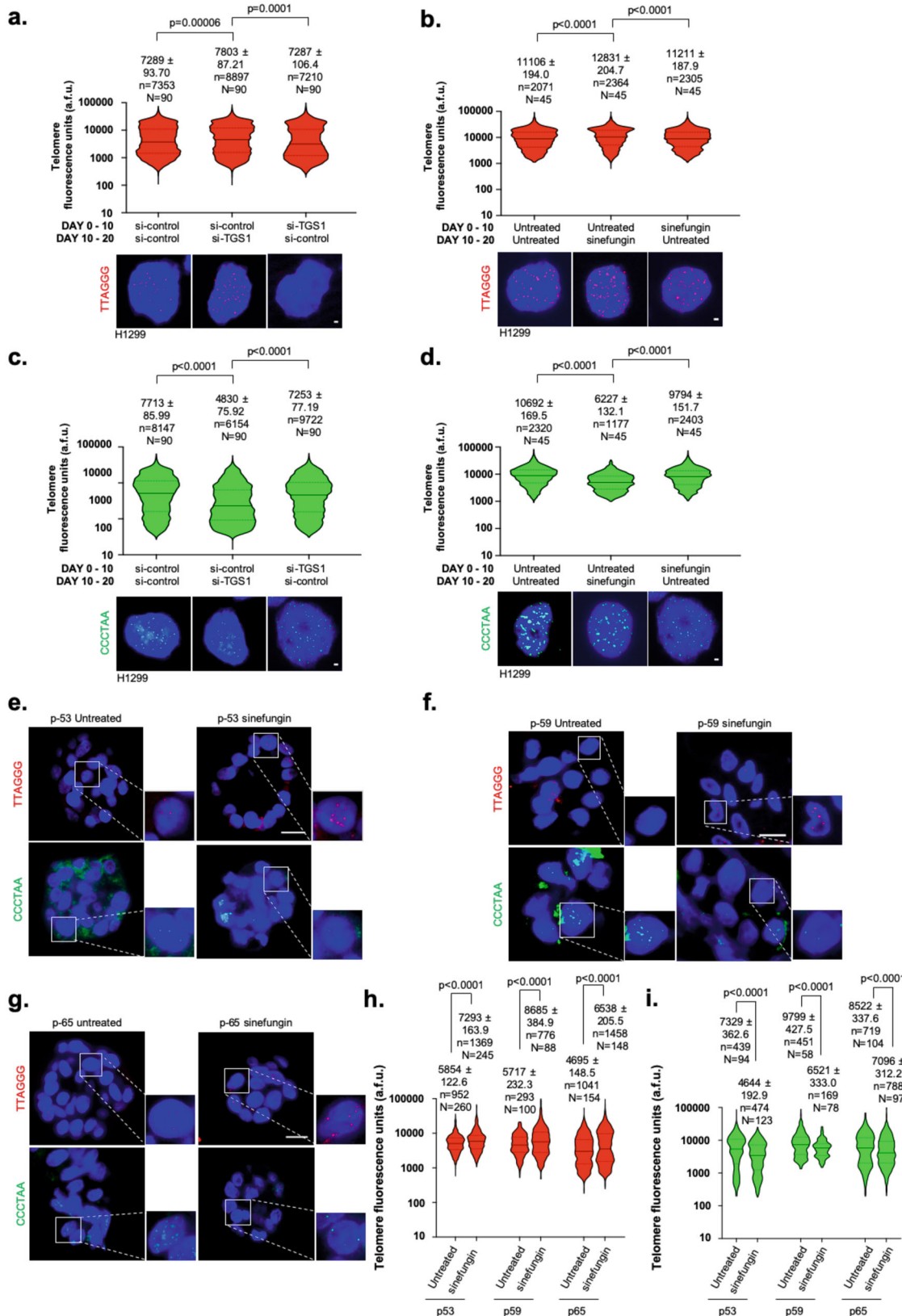

context, POLD3 is reported to stimulate Pol-δ activity and promotes ALT-mediated telomere extension[12,51]. We therefore set out to test whether POLD3 is associated with telomeres in sinefungin- and 5-Aza-2′-deoxycytidine-treated H1299 cells that display elongation of both telomere single strands. Colocalization events between POLD3 and TRF1 were low in control and sinefungin-treated cells, which is consistent with the lack of elongation of both telomere strands (Fig. 6o, p). However,

**Fig. 3 Loss of TGS1 results in telomeres with extended 3′ overhangs. a**, **c** Telomere DNA-FISH on interphase cells using telomere G-rich strand (red) or telomere C-rich strand- (green) specific probes after the indicated 20-day treatment scheme. DNA-FISH of all samples was performed at day 20. Top panels: telomere fluorescence intensity was quantified for each telomere G-rich strand signal (red) or C-rich telomere-strand signal (green). Bottom panel: representative images of DNA-FISH on G-rich telomere (red) or C-rich telomere (green) strands in H1299 cells following the indicated siRNA-transfection scheme. **b**, **d** Telomere DNA-FISH on interphase cells using G-rich telomere strand (red) or C-rich strand telomere- (green) specific probes after the indicated 20-day treatment scheme. DNA-FISH of all samples was performed at day 20. Top panels: telomere fluorescence intensity was quantified for each telomere G-rich strand signal (red) or C-rich telomere-strand signal (green). Bottom panel: representative images of DNA-FISH on G-rich telomere (red) or C-rich telomere (green) strands in H1299 cells following the indicated scheme for sinefungin treatment. **e–g** Representative DNA-FISH images of G-rich telomere (red) and C-rich telomere strands (green) on lung tumor organoids untreated or treated with sinefungin for 10 days. **h**, **i** Telomere fluorescence intensity of G-rich telomere (red) and C-rich telomere (green) strands of tumor organoids shown in **e–g**. For telomere-length measurements, violin-plot diagrams (**a**, **b**, **c**, **d**, **h**, **i**) are shown: middle line represents median arbitrary fluorescence units (a.f.u.), dotted lines mark the highest and lowest quartile. Mean a.f.u. values with standard deviation are indicated; $N$ = number of analyzed nuclei; $n$ = telomere-repeat signals. Red and green, telomere-strand-specific DNA-FISH probes, DAPI-stained nuclei. Scale bars in **a**, **b**, **c**, **d** correspond to 1 μm. Scale bars in **e–g** correspond to 10 μm. A two-tailed Student's $t$-test was used to calculate statistical significance; p-values are shown.

combining sinefungin and 5-Aza-2′-deoxycytidine treatment resulted in a significantly improved POLD3 recruitment to telomeres, suggesting ALT-mediated telomere extension. Together, our data indicate that sinefungin induced formation of recombinogenic telomere substrates and stimulation of the APB compartment by 5-Aza-2′-deoxycytidine translates into elongation of both telomere strands via the ALT pathway.

## Discussion

Here we show that TGS1-mediated 2,2,7-TMG cap formation at the 5′end of the hTR noncoding RNA has a critical role in directing the telomerase complex to chromosome ends and to engage Cajal bodies in telomere maintenance. 2,2,7-TMG capping increases hTR levels, but is not essential for the enzymatic activity of telomerase in vitro. Our study demonstrates that TGS1 depletion or inhibition by sinefungin blocks telomerase recruitment to telomeres, leading to EXO1-mediated formation of single-stranded 3′-telomere overhangs that engage in RAD51-dependent homologous recombination and APB formation in telomerase-positive H1299 cells and lung tumor organoids. Telomere single-strand alteration in H1299 TGS1-knockdown cells is linked with classic telomere features found in telomerase-negative ALT cells[52], such as increased DNA:RNA hybrid load, replication stress, recruitment of RAD51 and BLM to telomere repeats, APB formation, elevated telomere sister chromatid-exchange rates, and increase in C-circle abundance. These observations are in line with key features of break-induced DNA-replication (BIR) pathway that drives ALT[12,52]. This suggests that ALT features may have a critical role in supporting length alterations of the G-rich telomere strand observed by denaturing telomere-restriction fragment analysis in TGS1-knockout UMUC3 cells[27]. Our telomere-strand-specific DNA-FISH data show that acute depletion of TGS1 does not result in elongation of both telomere single strands in lung cancer cells. However, we demonstrate that stimulation of the APB compartment using 5-Aza-2′-deoxycytidine allowed to overcome this limitation and was linked with an increased recruitment of the DNA-polymerase δ-subunit POLD3 to telomeres, providing a rationale for the observed length increase of both telomere single strands.

Our data highlight a central role for TGS1-dependent 2,2,7-TMG capping of hTR to assure selective, telomerase-dependent telomere maintenance and restrict the formation of telomeric substrates that lead to aberrant ALT in telomerase-positive cancer cells. Inhibition of TGS1 function may represent an alternative strategy to block telomerase-dependent telomere maintenance of telomeres in cancer, but may also provide new inroads in improving telomere reserve in adult stem cells or patients with premature aging phenotypes due to reduced telomerase activity.

## Methods

**Cell lines and cell culture**. Human cancer cell lines were obtained from ATCC and have not been cultured for longer than 6 months. H1299 (carcinoma, non-small-cell lung cancer) cells were cultured in Roswell Park Memorial Institute (RPMI) medium (Euroclone) supplemented with 10% fetal bovine serum (Euroclone), 1% L-glutamine (Gibco), and 1% penicillin/streptomycin (Euroclone). U-2 OS (osteosarcoma) cells were cultured in low- glucose Dulbecco's modified Eagle's (DMEM) medium (Lonza) with 10% fetal bovine serum (Gibco), 1% L-glutamine (Gibco), and 1% penicillin/streptomycin (Gibco). H1299 cells stably overexpressing FLAG-hTERT and hTR were generated through the transfection of the respective linearized vector and were selected using blasticidin (5 mg/mL) and puromycin (1 mg/mL).

**Generation of lung tumor organoids**. The research protocol was approved by the ethics committee of the region Friuli Venezia Giulia, Italy (comitato etico unico regionale, prot. n. 5798, 22.02.2018, Aviano, Italy). Written informed consent was obtained from all participants of the study; enrolled patients did not receive therapeutic treatment prior to surgical resection of the tumor and were not selected based on mutation profiles. Experimental procedures were conducted in compliance with the institutional guidelines. Samples were confirmed as tumor or normal tissue on the basis of histopathological assessment. Punch biopsies (~1–4 cm³) from surgically resected primary lung adenocarcinomas were used to generate tumor organoids as previously described[53]. Briefly, human biopsies were washed three times with ice-cold 1x PBS and sectioned with sterile blades. Samples were incubated with 1,5 mg/mL collagenase (Sigma-Aldrich) in AdDF+++ medium (DMEM/F12 medium (Lonza) supplemented with glutamine 500 μg/mL, Primocin 50 μg/mL, 200 mg/mL streptomycin, and Hepes 10 mM) at 37 °C for 2 h under gentle agitation. Subsequently, AdDF+++ supplemented with 10% FBS was used to neutralize collagenase and was passed through a 100-μm cell strainer (BD Falcon, CA, USA). Cells were centrifuged at 300 × g for 5 min at room temperature. In case of a visible red pellet, erythrocytes were lysed in a 4:1 mixture of ammonium chloride and cold-modified HBSS Hank's Balanced Salt Solution (Sigma-Aldrich) for 3 min on ice before the addition of AdDF+++ and centrifuged at 300 × g for 5 min at room temperature. The pellet was then resuspended in Matrigel (Growth Factor Reduced, BD) and 40 μl of drops of Matrigel-cell suspension were allowed to solidify on prewarmed 24-well suspension culture plates at 37 °C for 10–20 min. After solidification, 400 μl of AdDF+++ supplemented with Noggin-conditioned medium 25% v/v, R-Spondin 1 conditioned medium 25% v/v, FGF7 25 ng/mL (Peprotech), FGF10 100 ng/mL (Peprotech), ALK inhibitor 500 nM (Tocris), 5 μM ROCK inhibitor (Abmole), SB202190 500 μM (Sigma-Aldrich), B27 (Invitrogen), N-Acetylcysteine 125 mM (Sigma-Aldrich), and Nicotinamide 5 mM (Sigma-Aldrich), was added to each well and plates transferred to humidified 37 °C/5% CO₂ incubators at ambient O₂. The medium was changed every 4 days, and organoids were passaged after 1–3 weeks. For passaging, organoids were resuspended in cold 1X PBS and mechanically sheared using flamed glass Pasteur pipettes. Organoids were then gently resuspended in Dispase (Stemcell) 5 U/mL in AdDF+++ medium for 10 min at 37 °C in order to digest the remaining Matrigel. AdDF+++ supplemented with 10% FBS was added to the pellet and centrifuged at 300 × g for 5 min at 2–8 °C. The cell pellet was washed by gentle resuspension in cold 1X PBS and centrifuged at 2–8 °C. Dense organoids were dissociated by resuspension in TrypLE Express (Invitrogen), incubation for 1–5 min at room temperature, and mechanical shearing using flamed glass Pasteur pipettes. AdDF+++ was added and cells were concentrated by centrifugation at 300 × g. Organoid fragments were resuspended in cold Matrigel and reseeded (ratio 1:1–1:4).

**Whole-exome sequencing and mutation calling**. DNA extracted from the matched tumor organoids, primary tumor, and normal tissues was used to construct

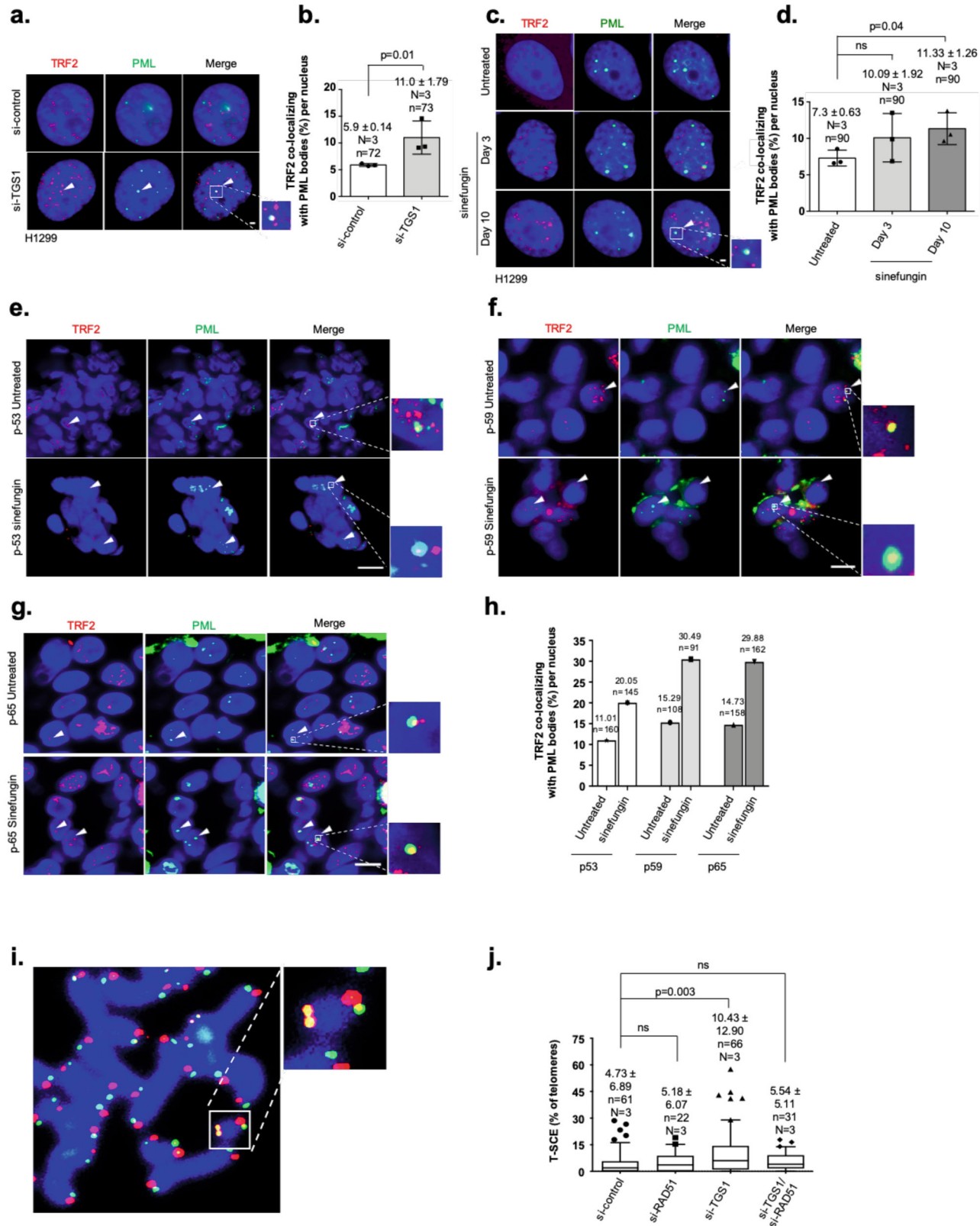

2 × 150-bp libraries, with the SureSelect Human All Exon V6 kit, and subjected to whole- exome sequencing (WES) on the Illumina NovaSeq platform. Quality checks for raw reads were performed using FastQC (http://www.bioinformatics.babraham.ac.uk/projects/fastqc). High-quality reads were aligned to the GRCh37 human reference genome by BWA[54] after adapter and PCR duplicate removal by the Trimmomatic[55] and MarkDuplicates module from Picard tools (Broad

Institute: Picard:A set of command-line tools (in Java) for manipulating high-throughput sequencing (HTS) data and formats such as SAM/BAM/CRAM and VCF. http://broadinstitute.github.io/picard/), respectively. Deduplicated reads were realigned at known indel positions and systematic sequencing errors fixed with recalibration by GATK (https://software.broadinstitute.org/gatk/best-practices). The somatic single- nucleotide variants (SNVs) and short indels were identified in

**Fig. 4 Loss of TGS1 induces RAD51-dependent telomere recombination and APB formation. a** Representative images of combined immunofluorescence using specific anti-TRF2 and anti-PML antibodies in H1299 cells repeatedly transfected with indicated siRNAs for 10 days. **b** Percentage of PML bodies colocalizing with telomeres (TRF2) per nucleus of experiment shown in **a**. **c** Representative images of combined immunofluorescence with anti-TRF2 and anti-PML antibodies in H1299 cells untreated or treated with sinefungin for 3 and 10 days. **d** Percentage of PML bodies colocalizing with telomeres (TRF2) per nucleus of experiment shown in **c**. **e–g** Representative images of combined immunofluorescence using anti-TRF2 and anti-PML antibodies on tumor lung organoids obtained from different patients. Lung tumor organoids were untreated or treated with sinefungin for 10 days. **h** Percentage of PML bodies colocalizing with telomeres (TRF2) in lung tumor organoid cells. **i** Representative image of a metaphase spread showing telomeric sister chromatid exchange (T-SCE), as detected by CO-FISH (shown as yellow signals). **j** Quantification of T-SCE events in H1299 cells transfected with indicated siRNAs. For quantifications shown in **b**, **d**, and **j**, mean values are indicated; whiskers indicate standard deviation. For box-plot diagram (**j**): middle line represents median; boxes extend from the 25th to 75th percentiles. The whiskers mark the 10th and 90th percentiles. Mean T-SCE values and standard deviation are indicated. $N$ = number of independent experiments. $n$ = number of analyzed nuclei/metaphase spreads. Bars in panels **b**, **d** indicate mean values. Red and green, epitopes detected by immunofluorescence; blue, DAPI-stained nuclei. Scale bars in a, c correspond to 1 μm. Scale bars in **e–g** correspond to 10 μm. A two-tailed Student's $t$-test was used to calculate statistical significance; $p$-values are shown.

tumor tissues and organoids versus the matched normal tissues by integrating MuTect2[56] and VarScan2[57] algorithms, with default parameters. Somatic variants were annotated using Annovar[58].

**siRNA transient transfection**. For siRNA transfection, RNAi-MAX Lipofectamine (Invitrogen) was used according to the manufacturer's suggestions. siRNAs have been transfected at a final concentration of 30 nM for 72 h. For plasmid transfections, Lipofectamine 2000 (Invitrogen) was used according to the manufacturer's suggestions. siRNAs used are listed in Supplementary Table 1.

**Immunohistochemistry**. Tissues dedicated to immunohistochemistry were fixed in 10% buffered formalin. After dehydration, samples were embedded in paraffin and 3-μm sections prepared. Organoids were released from Matrigel and fixed in 4% buffered paraformaldehyde, washed twice in phosphate-buffered saline (PBS), and resuspended in predissolved 1% agarose. Agarose blocks were embedded in paraffin and 3-μm sections prepared. Antigen retrieval was performed using cell phosphate buffer solution at 95 °C for 40 min. Endogenous peroxidase was inhibited with $H_2O_2$ at 3% (Bioptica) for 10 min. Samples were incubated with a rabbit monoclonal anti-CK7 primary antibody (clone SP52, Roche-Ventana) for 32 min at 36 °C. An UltraView Universal DAB Detection Kit (Roche) was used for visualization of IHC signals. For standard H&E staining, slides were deparaffinized in xylene, rehydrated by graded alcohol, and stained with hematoxylin and eosin to appreciate the cellular and tissue-structure details. Images were acquired on a D-sight brightfield slide scanner (Menarini Diagnostics) and analyzed by lung cancer pathologists.

**Immunofluorescence**. Cells were fixed in 4% paraformaldehyde (PFA) for 15 min, treated with sodium citrate buffer [0.1% (w/v), 0.5% Triton X-100] for 5 min. Slides were blocked for 1 h in 3% BSA, 0.1% Tween-20 in 1X PBS (blocking solution), and incubated with primary antibodies diluted in blocking solution for 2 h at room temperature.

Slides were washed twice in 0.3% BSA, 0.1% Tween-20 in 1X PBS (washing solution), and incubated with secondary antibodies diluted in washing solution at room temperature. Slides were washed once in washing solution, once in 0.1% Tween-20 1X PBS, including DAPI (1 μg/mL, Sigma), and mounted in ProLong™ Gold Antifade Mountant (ThermoFisher Scientific). For S9.6 immunofluorescence, cells were fixed and permeabilized with ice-cold methanol for 10 min and acetone for 1 min on ice as previously described[49]. Blocking, antibody, and washing solutions were performed in 4X SSC. After fixation, cells were treated with RNase H1 (New England Biolabs) for 1 h at 37 °C, according to the manufacturer's instructions. Cells with at least 5 nuclear extranucleolar S9.6 foci were analyzed.

For organoid immunofluorescence experiments, organoids were fixed for 20 min in 4% PFA and resuspended in 1% agarose. Agarose plugs were embedded in paraffin blocks and then cut into thin slices (4–6 μm). To remove paraffin, sections were washed 3x in xylene for 5 min. Samples were rehydrated by washing in 100%, 90%, and 70% ethanol for 5 min and rinsed with $H_2O$. For antigen unmasking, slides were boiled at 121 °C for 30 min in 10 mM sodium citrate buffer, pH 6.0. Sections were rinsed in $H_2O$ and washed twice in 0.2% Tween-20 in 1X PBS (washing solution) for 5 min each. Cells were then blocked for 20 min at room temperature in 5% nonfat dry milk in 1X PBS Tween-20. Sections were incubated overnight at 4 °C with primary antibodies (as indicated in Supplementary Table 2) diluted in blocking solution (5% nonfat dry milk in PBS, 0,2% Tween-20). Slides were washed three times for 5 min in washing solution and incubated with secondary antibodies (as indicated in Supplementary Table 2) diluted in blocking solution for 30 min at room temperature. Then slides were washed twice in washing solution for 5 min and once for 5 min in blocking solution, including DAPI (1 μg/mL, Sigma). Samples were mounted in ProLong™ Gold Antifade Mountant (ThermoFisher Scientific). Primary and secondary antibodies are listed in Supplementary Table 2. Colocalization events were quantified using ImageJ

1.46r or by visual inspection. A two-tailed Student t-test was used to calculate statistical significance.

**Interphase telomere DNA-FISH**. Interphase DNA-FISH was performed as previously described[59]. Briefly, cells were fixed in 4% PFA for 20 min. After three washes with 1X PBS for 5 min, slides were dehydrated by washing in 70%, 90%, and 100% ethanol for 5 min each. Slides were allowed to dry for 15 min at room temperature. A Cy3-labeled $(CCCTAA)_3$ and Alexa-488-labeled $(TTAGGG)_3$ probes were added to the sample, and after denaturation at 80 °C for 3 min, slides were incubated for 2 h at room temperature in a humid chamber in the dark. Subsequently, slides were washed twice with FISH solution (70% formamide, 10 mM Tris, pH 7.2, and 0.1% BSA) under agitation for 15 min at room temperature and three times with 0.01% Tween-20 in 1X TBS at room temperature. DNA was stained with DAPI (1 μg/mL in 0.01% Tween-20, Sigma). Slides were dehydrated by washing in 70%, 90, and 100% ethanol. Samples were mounted with ProLong™ Gold Antifade Mountant (ThermoFisher Scientific). Interphase nuclei were analyzed using spot IOD analysis (TFL-TELO) software. A two-tailed Student's t-test was used to calculate statistical significance.

For organoid-interphase telomere DNA-FISH experiments, a pellet of organoids was fixed for 20 min in 4% PFA and processed to 1% agarose plus. Sections (4–6 μm) were used to perform DNA-FISH experiments. Slides were washed three times for 5 min each in xylene to remove paraffin. Cells were then rehydrated by washing in 100%, 90%, and 70% ethanol for 5 min each, and then rinsed with $H_2O$. Slides were washed in 1X PBS for 15 min at room temperature, fixed in formaldehyde 4% in 1X PBS for 2 min, and washed three times for 5 min in 1X PBS. Organoids were then digested in Pepsin/HCl at 37 °C for 10 min. After three washes with 1X PBS for 5 min, slides were dehydrated by washing in 70%, 90%, and 100% ethanol for 5 min. Slides were allowed to dry for 15 min at room temperature. Samples were then incubated with a Cy3-labeled $(CCCTAA)_3$ and Alexa-488-labeled $(TTAGGG)_3$ probes and then processed following the DNA-FISH protocol for interphase cells.

**Chromosome-orientation FISH (CO-FISH)**. Cells at 70% confluence were cultivated in the presence of 5′-bromo-2′-deoxyuridine (BrdU, 10 μM, Sigma) for 18–24 h. Colcemid was added to cells at a final concentration of 1 μg/mL for 2 h.

Cells were recovered and metaphase spreads were prepared as previously described[59]. CO-FISH was performed as previously described using a Cy3-$(CCCTAA)_3$ FISH probe followed by a $(TTAGGG)_3$ FISH probe labeled with Alexa-488[60,61].

**RNA extraction and RT-PCR**. Total RNA was prepared using QIAzol Lysis Reagent (Qiagen) and reverse-transcribed using the QuantiTect Reverse Transcription Kit (Qiagen) according to the manufacturer's instructions. Quantitative RT-PCR was performed using a SYBR green real-time PCR mix (Applied Biosystem) in a StepOnePlus real-time PCR machine (Applied Biosystem). For organoid samples, organoids were extracted from Matrigel by incubating samples for 10 min at 37 °C in Dispase solution 5 U/mL (StemCell). Recovered organoids were then washed with 1X PBS, recovered by centrifugation at 300xg for 5 min at room temperature. Total RNA from organoids was then prepared using QIAzol Lysis Reagent (Qiagen) and reverse-transcribed with QuantiTect Reverse Transcription Kit (Qiagen) according to the manufacturer's instructions. PCR primers used for quantitative real-time PCR are listed in Supplementary Table 3.

**TRAP assay**. Telomerase activity was measured using TRAPeze Telomerase Detection Kit (Millipore S7700) following the manufacturer's instructions. Cells were washed with ice-cold 1X PBS and lysed with 1X CHAPS buffer for 30 min on ice. About 100, 50, or 25 ng of lysate was subjected to the TRAP assay using DreamTaq DNA Polymerase (ThermoScientific). Heat-inactivated lysates and

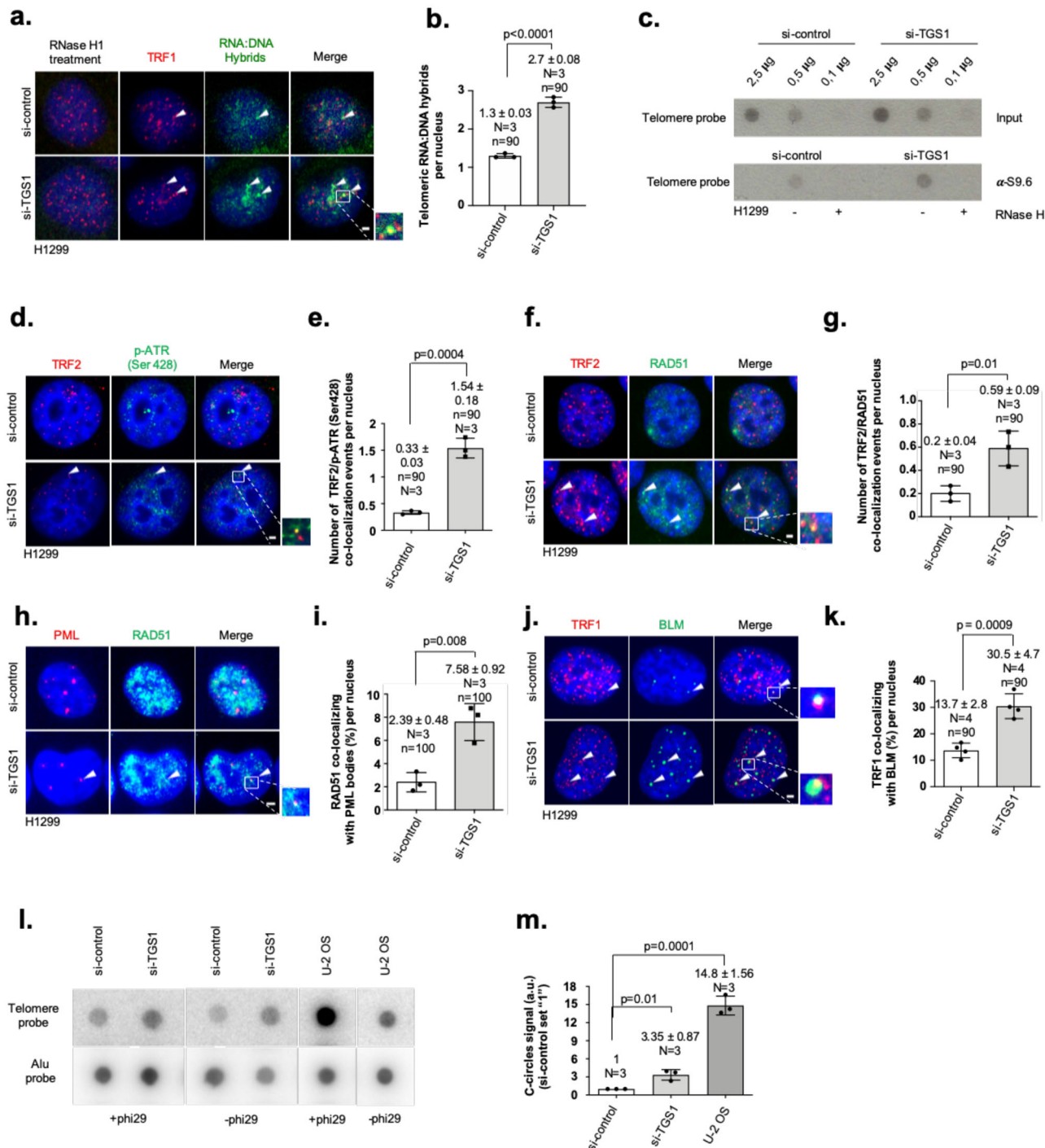

TSR8 were used as control. Samples were then run on a 10% nondenaturing polyacrylamide gel in 0,5X TBE buffer.

Subsequently, the gel was stained with SYBR Gold nucleic acid gel (Life Technologies) according to the manufacturer's instructions. The acquisition of images was performed using a gel DOC XR system (BIO-RAD). ImageJ Software was used to determine the intensity of the TRAP products. Intensity signal of non-heat-treated samples ($x$), heat-treated samples ($x_0$), primer–dimer/PCR contamination control ($r_0$), positive PCR control TSR8 ($r$), and S-IC (standard internal PCR control) in non-heat-treated samples ($c$) and TSR8 control ($c_r$) was measured. The following formula was used to calculate TRAP activity: $(x–x_0)/c(r–r_0)/c_r*100$.

**Western blot**. Whole-cell lysates were prepared as previously described[62]. Briefly, whole-cell lysates were prepared using a modified RIPA buffer (20 mM Tris-HCl, pH 7.5, 350 mM NaCl, 1 mM EDTA, 1 mM EGTA, 1% NP-40, 1% sodium

deoxycholate, 2.5 mM sodium pyrophosphate, 1 mM β-glycerophosphate, 1 mM $Na_3VO_4$, and 1 mg/mL leupeptin). Samples were supplemented with complete protease inhibitor (Roche), sonicated, and cleared by centrifugation at 4 °C. Supernatants were recovered and used for Western blotting according to standard procedures. For organoid samples, organoids were extracted from Matrigel using Dispase solution 5 U/mL (StemCell) for 10 min at 37 °C. Recovered organoids were then washed with 1X PBS and recovered by centrifugation at 300xg for 5 min. Whole-organoid lysates were resuspended in RIPA buffer and processed as previously described. Primary antibodies are listed in Supplementary Table 2.

**Telomere ChIP assays**. Chromatin immunoprecipitation of telomere was performed as described[63]. In total, $6 \times 10^6$ cells were cross-linked for 15′ by adding formaldehyde directly to culture medium to a final concentration of 1%. Glycine was added to a final concentration of 0.125 M to stop the cross-linking. Cells were then collected and lysed in lysis buffer (0.5 mM EDTA, 50 mM Tris-HCl, pH 8, and

**Fig. 5 TGS1 suppresses ALT in telomerase-positive cancer cells. a** Representative images of combined immunofluorescence using anti-TRF1 and monoclonal anti-DNA:RNA hybrid-specific antibodies (S9.6) in H1299 cells transfected with indicated siRNAs. RNase H1-treated cells were used as negative control. RNase H1 treatment validates S9.6 antibody specificity. **b** Quantification of DNA:RNA hybrids/TRF1 colocalization events per nucleus of experiment shown in **a**. **c** DNA:RNA- immunoprecipitation experiment (DRIP) using H1299 cells. A S9.6 monoclonal anti-DNA:RNA hybrid antibody was used for immunoprecipitation. Serial dilutions of genomic material (input) prepared from H1299 cells were loaded. **d** Representative images of combined immunofluorescence using specific anti-TRF2 and anti-phospho ATR (pATR-Ser 428) antibodies on H1299 cells transfected with indicated siRNAs. **e** Quantification of colocalization events per nucleus of experiment shown in **d**. **f** Representative images of combined immunofluorescence with anti-TRF2 and anti-RAD51-specific antibodies on H1299 cells transfected with indicated siRNAs. **g** Quantification of TRF2/RAD51 colocalization events per nucleus of experiment shown in **f**. **h** Representative images of combined immunofluorescence with anti-PML and anti-RAD51 antibodies in H1299 cells transfected with indicated siRNAs. **i** Percentage of PML bodies colocalizing with RAD51 per nucleus of experiment shown in **h**. **j** Representative images of combined immunofluorescence with anti-TRF1 and anti-BLM-specific antibodies in H1299 cells transfected with indicated siRNAs. **k** Percentage of BLM foci colocalizing with TRF1 per nucleus of experiment shown in **j**. **l** C-circle assay experiment using H1299 cells transfected with the indicated siRNAs. No phi29 treatment and U-2 OS were used as negative and positive controls, respectively. Membranes were probed with a radiolabeled telomere probe and a human Alu-repeat probe used as loading control. **m** Quantification of C-circle assay experiments shown in **l**. Bars in panels **b**, **e**, **g**, **i**, **k**, and m indicate mean values; whiskers indicate standard deviation; $N$ = number of independent experiments. $n$ = number of analyzed nuclei. Arrowheads indicate colocalization events. Red and green, epitopes detected by immunofluorescence; blue, DAPI-stained nuclei. Scale bar, 1 μm. A two-tailed Student's $t$-test was used to calculate statistical significance; $p$-values are shown. Uncropped blots in Source Data.

1% SDS) containing protease inhibitors. Lysates were sonicated to obtain chromatin fragments with 500–800-bp length and were precleared for 1 h at 4 °C with protein-A or -G Agarose beads. Chromatin was immunoprecipitated overnight at 4 °C on a rotating wheel using 2,5 μg of primary antibodies (Supplementary Table 2) or mouse control IgGs. DNA was purified from immunoprecipitated chromatin by RNaseA and Proteinase-K digestion, phenol–chloroform extraction, followed by ethanol precipitation. DNA was transferred to a Hybond N+ membrane and hybridized with a dCTP-[α-32P] body-labeled telomere-repeat probe using the Random primers DNA labelling system (Invitrogen).

**RNA immunoprecipitation**. Total RNA extracted from cells was resuspended in RIPA buffer (50 mM Tris-HCl, pH 7.5, 150 mM NaCl, 1% Triton X-100, 1% sodium deoxycholate, and 0.1% SDS) containing RNase inhibitors (RNaseOUT-Invitrogen). About 50 μg of total RNA was incubated with 5 μg of anti-2,2,7-trimethylguanosine antibody (Supplementary Table 2) for 4 h at 4 °C. Protein-G Agarose beads were then added and the incubation continued for 3 h. After three washes of 5′ with RIPA supplemented with RNaseOUT, the pelleted resin was centrifuged and the bounded RNA was extracted using QIAzol Lysis Reagent (Qiagen) according to the manufacturer's instructions. The purified RNA was reverse- transcribed using QuantiTect Reverse Transcription Kit (Qiagen), which employs random oligo amplification, and quantitative PCR was performed using the SYBR Green Master Mix (Applied Biosystem) and analyzed with a StepOnePlus real-time PCR machine (Applied Biosystems).

**Northern blot**. Total RNA (10 μg) was loaded onto 1.2% formaldehyde agarose gel (20 mM MOPS, 1.1% formaldehyde) and separated by electrophoresis in a buffer of 20 mM MOPS. RNA was then transferred to Hybond N+ membrane and hybridized overnight at 65 °C in hybridization buffer (7% SDS, 250 mM NaPO4, pH 7.2) containing a dCTP-[α-32P]-labeled hTR or GAPDH probe using the Random primers DNA labelling system (Invitrogen). Afterward, membranes were washed twice for 20′ at 65 °C with Wash Buffer (1% SDS, 20 mM NaPO4, pH 7.2). Specific bands were detected by exposing the membrane to autoradiography films (GE Healthcare).

**Immunofluorescence combined with RNA-FISH**. Cells were rinsed once with 1X PBS and fixed for 10 min with 4% formaldehyde, 10% acetic acid, and 1X PBS for 10 min at room temperature. After two washes with 1X PBS, cells were permeabilized with 70% ethanol overnight at 4 °C. Cells were rehydrated for 5 min at room temperature in 2X SSC and 50% formamide for 1 h at 37 °C. Cells were then hybridized for 4 h with a Cy3-labeled hTR probe (FISH Tag DNA kit, Invitrogen) in a humid chamber (2X SSC, 50% formamide) at 37 °C. After hybridization, cells were washed once in Tris-HCl, pH 7.4; 0,15 M NaCl; and 0,05% Tween-20 for 30 seconds at room temperature, and three times in 2X SSC for 5 min at 37 °C. Subsequently, cells were washed twice with 1X PBS for 10 min at room temperature. Cells were blocked for 30 min in 3% BSA, 0.1% Tween-20 in 1X PBS (blocking solution), and incubated with primary antibody diluted in blocking solution for 2 h at room temperature. Cells were then washed twice in 0,3% BSA, 0,1% Tween-20 in 1X PBS for 10 min at room temperature. Subsequently, samples were incubated with secondary antibodies for 1 h at room temperature. After a 5-min wash in 1X PBS containing 1 μg/mL DAPI, slides were rinsed in 4X SSC and mounted in ProLong™ Gold Antifade Mountant (ThermoFisher Scientific). All antibodies were diluted in 3% BSA, 0,1% Tween-20 in 1X PBS. Primary and secondary antibodies are listed in Supplementary Table 2.

**C-circle assay**. C-circle assay was performed as previously described[64]. Briefly, 25 or 150 ng of HinfI- and RsaI- digested genomic DNA was incubated with phi29 polymerase (New England B; 0.75 U phi29 polymerase, 0.2 mg/mL BSA, 0.1% Tween-20, 1 mM dNTP mix, and 1X phi29 buffer) at 30 °C for 4 h and then at 70 °C for 20 min. Samples were spotted on a Hybond N+ membrane and hybridized with a dCTP-[α-32P]-labeled telomere-repeat probe using a Random primer DNA labelling system according to the manufacturer's suggestions (Invitrogen), in native conditions. A human Alu probe was used as loading control. The intensity of each sample was calculated by densitometric analysis of autoradiographs using ImageJ software. Briefly, any global background controls from all samples were subtracted. Each sample was corrected for loading by dividing the amount of Alu repeats detected. Finally, from any sample previously normalized for loading, specific normalized background controls (no Phi control) were subtracted.

**G-strand overhang assay**. G-strand overhang analyses were performed with modifications[65]. Briefly, genomic DNA was purified using Quick-DNA Miniprep Kit (Zymo Research) following the manufacturer's instructions. About 4 μg of genomic DNA were digested with 30U Hinf/RsaI (New England Biolabs) overnight at 37 °C. Digested genomic DNA samples were run on a 0.8% agarose gel (24 h, 1 V/cm) in 1X TAE buffer (40 mM Tris-HCl, 20 mM acetic acid, and 1 mM EDTA, pH 8.0). Samples were transferred in 2X SSC overnight to a Hybond-N+ nylon membrane and hybridized as previously described[59]. A human COT-1 DNA probe was used as loading control. The intensity of the obtained band on autoradiographs was quantified by densitometric analysis using the ImageJ software.

**DNA:RNA immunoprecipitation (DRIP)**. Cells harvested by scraping were lysed overnight at 37 °C and for 4 h at 65 °C at 850 rpm in Lysis buffer (1% SDS, 20 mM Tris-HCl, pH 7.5, 40 mM EDTA, pH 8, and 100 mM NaCl) and TE buffer (100 mM Tris-HCl, pH 8, 10 mM EDTA, pH 8) supplemented with Proteinase K (150 μg/mL). Nucleic acids were extracted with phenol/chloroform/isoamyl alcohol (25:24:1), followed by ethanol precipitation and resuspended in Elution buffer (5 mM Tris-HCl, pH 8.5). Purified nucleic acid samples were sonicated and 6 μg was treated with 24 U of RNase H (New England Biolabs) overnight at 37 °C. Dynabeads Protein A (Life Technologies) were preblocked for 2 h at 4 °C in 1X PBS supplemented with 5 mM EDTA and 0.5% BSA. Blocked Dynabeads Protein A were washed and resuspended in 1 mL of IP buffer (50 mM Hepes/KOH, pH 7.5, 0.14 M NaCl, 5 mM EDTA, pH 8, 1% Triton X-100, and 0.1% Na- deoxycholate) and incubated with 3 μg of S9.6 antibody for 4 h at 4 °C on a rotating wheel. About 6 μg of sonicated samples untreated or treated with RNase H were added to the solution containing the S9.6 antibody and the Dynabeads Protein A and incubated overnight at 4 °C on a rotating wheel.

Beads were washed one time with IP buffer, Low Salt buffer (50 mM Hepes/KOH, pH 7.5, 0.15 M NaCl, 5 mM EDTA, pH 8, 1% Triton X-100, and 0.1% Na-deoxycholate), and High Salt buffer (50 mM Hepes/KOH pH 7.5, 0.5 M NaCl, 5 mM EDTA, pH 8, 1% Triton X-100, and 0.1% Na-deoxycholate), and two times with TE buffer. Elution was performed with 100 μl of Elution buffer (50 mM Tris-HCl, pH 8, 10 mM EDTA pH 8, and 1% SDS) at 65 °C for 15 min. The supernatant was recovered, purified with QIAquick PCR Purification Kit (Qiagen). DNA was spotted on a Hybond N+ membrane and hybridized with a dCTP-[α-32P] body-labeled telomere-repeat probe using the random primer DNA labelling system (Invitrogen).

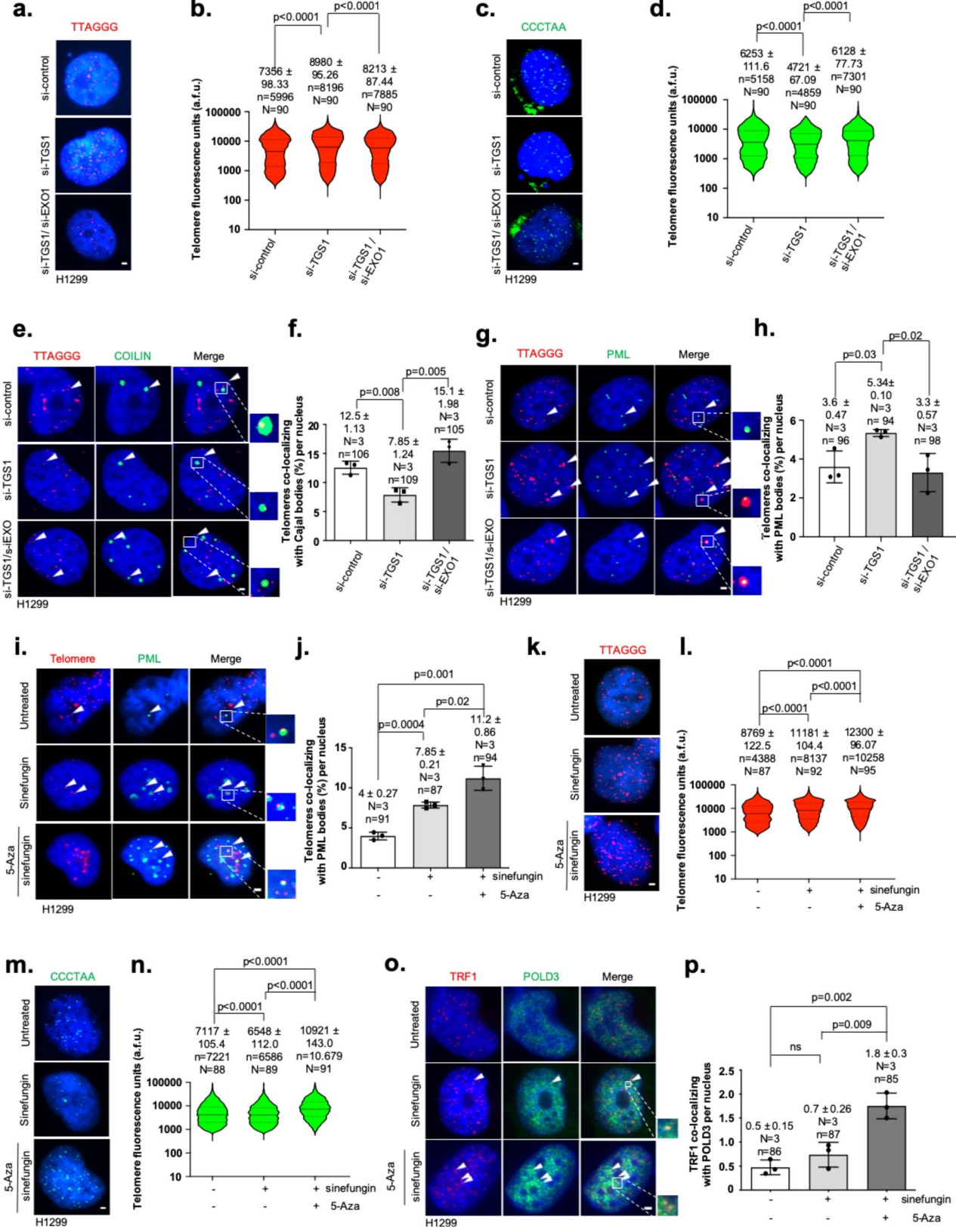

**Fig. 6 TGS1 regulates EXO1 activity at telomeres. a** Representative images of DNA-FISH experiment using a telomere G-rich strand-specific probe on H1299 cells transfected, as described, for 10 days. **b** Quantification of telomere G-rich strand fluorescence intensity of cells in **a. c** Representative images of DNA-FISH experiment using telomere C-rich strand-specific probe on H1299 cells transfected, as indicated, for 10 days. **d** Quantification of telomere C-rich strand fluorescence intensity of cells in **c. e** Representative images of telomere DNA-FISH combined with immunofluorescence using anti-Coilin antibody in H1299 cells transfected with indicated siRNAs for 10 days. **f** Quantification of experiment shown in **e. g** Representative images of telomere DNA-FISH combined with immunofluorescence using anti-PML antibody in H1299 cells transfected with indicated siRNAs for 10 days. **h** Quantification of experiment shown in **g. i** Representative images of telomere DNA-FISH combined with immunofluorescence using anti-PML antibody in H1299 cells treated as indicated. **j** Quantification of experiment shown in **i. k** Representative images of DNA-FISH of G-rich telomere strands on H1299 cells treated as indicated. **l** Telomere fluorescence-intensity analysis of cells shown in **k. m** Representative images of DNA-FISH of C-rich telomere strands on H1299 cells treated as indicated for 10 days. **n** Telomere fluorescence- intensity analysis of cells shown in **m. o** Representative images of immunofluorescence analysis using anti-TRF1 and anti-POLD3-specific antibodies. H1299 cells were treated as indicated. **p** Quantification of experiment shown in **o**. In Fig. **b, d, l, n** violin-plot diagrams are shown: middle line represents median arbitrary fluorescence units (a.f.u.), dotted lines mark the highest and lowest quartile. For quantifications in **f, h, j, l, p** mean values are indicated; whiskers indicate standard deviation. Bars indicate mean values. In Fig. **b, d, l, n** $N$ = number of analyzed nuclei, $n$ = telomere-repeat signals. In Fig. **f, h, j, p** $N$ = number of independent experiments. $n$ = number of analyzed nuclei. Arrowheads indicate colocalization events. Red and green, epitopes detected by immunofluorescence; blue, DAPI-stained nuclei. Scale bar, 1 μm. A two-tailed Student's t-test was used to calculate statistical significance; *p*-values are shown.

**Reporting summary**. Further information on research design is available in the Nature Research Reporting Summary linked to this article.

## Data availability

The data supporting the findings of this study are available from the corresponding authors upon reasonable request. Whole-exome sequencing data have been deposited in the European Nucleotide Archive (ENA) at EMBL-EBI under accession number PRJEB51154. Source data for the figures and supplementary figures are provided as a Source Data file. Source data are provided with this paper.

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

## Acknowledgements

S.S. is supported by the Italian Association for Cancer Research (AIRC)—Investigator Grants 18381 and 23074 and the European Union, European Regional Development Fund, and Interreg V-A Italia-Austria 2014–2020 (project codes ITAT1096-P and ITAT1050). R.B is supported by the Italian Association for Cancer Research (AIRC)—Investigator Grant 17756. G.D.R is supported by a Telethon GPP13147 grant. S.C. is supported by a ASI 2016-6-U0 grant. G.D.S is supported by the European Union, European Regional Development Fund, and Interreg V-A Italia-Austria 2014–2020 (project code ITAT1096-P and project code ITAT1050-P-CARE); by the Italian University and Research Ministry (PRIN 2017HWTP2K_004 and MIUR-ARS01_00876-BIO-D), Italian Association for Cancer Research (AIRC) Special Program 5 × 1000 (22759), AIRC-IG (22174), and Ministero della Salute (RF-2019-12368718), A.Z. is supported by a post-doctoral fellowship from the Fondazione Veronesi. High- performance computing resources and support was provided via a CINECA award under the ISCRA initiative. F. d'A.d.F aboratory is supported by ERC advanced grant (TELORNAGING—835103); AIRC-IG (21762); Telethon (GGP17111); AIRC 5 × 1000 (21091); ERC PoC grant (FIREQUENCER— 875139); Progetti di Ricerca di Interesse Nazionale (PRIN) 2015 "ATR and ATM-mediated control of chromosome integrity and cell plasticity"; Progetti di Ricerca di Interesse Nazionale (PRIN) 2017 "RNA and genome Instability"; Progetto AriSLA 2021 "DDR & ALS"; POR FESR 2014–2020 Regione Lombardia (InterSLA project); FRRB—Fondazione Regionale per la Ricerca Biomedica—under the frame of EJP RD, the European Joint Programme on Rare Diseases with funding from the European Union's Horizon 2020 research and innovation programme under the EJP RD COFUND-EJP N° 825575.

## Author contributions

V.B. and O.S. performed the majority of the experiments and analyzed related data; M.S., P.V.B., S.K., A.Z., and G.D. performed experiments; M.S. performed exome-sequencing data analysis and prepared related manuscript text; D.B. tumor specimen preparation, IHC data interpretation; C.B. patent specimen management and immunohistochemistry; G.D.R. advises during the study; R.B. data interpretation and paper preparation; M.C. provided patient specimen and collection of informed patient consent; F.d'A.d.F, data interpretation and editing of the paper; F.Z. supervision of clinical study, data interpretation; G.D.S. advice, discussion of results of the study; S.S. conceived and supervised the study, interpreted data, and wrote the paper.

## Competing interests

G:D.R, S.C, and S.S. are inventors on patent applications related to chemical inhibition of TGS1 in telomere diseases ("Pharmaceutical composition for the chemical inhibition of TGS1 as therapeutic treatment for telomeropathies"; PCT/IB2021/054484). Other authors declare no competing interest.
