## [Peer Review File · Nature Communications]

TGS1 mediates 2,2,7-trimethylguanosine capping of the human telomerase RNA to direct telomerase dependent telomere maintenanceREVIEWER COMMENTS

Reviewer #1 (Remarks to the Author):

In this study, Buemi and co-workers investigate the role of Trimethylguanosine Synthetase 1 (TGS1) in telomerase-dependent telomere maintenance mechanisms (TMM) in human cells. Through a series of well-designed and present experiments, the authors convincingly demonstrate that TGS1 is important for TMM in human cells. They show that TGS1 facilitates the formation of a 2,2,7-TMG cap at the 5' end of telomerase hTR non-coding RNA, which directs the telomerase holoenzyme to Cajal bodies and chromosome ends to allow product TMM. In support of this conclusion, depletion of TGS1 or inhibition by Sinefungin blocks telomerase recruitment to telomeres but does not impact of enzymatic activity per se. In the absence/inhibition of TGS1, telomere ends are subject to ExoI dependent resection, resulting in increased 3' single stranded DNA, which engaged in Rad51 dependent homologous recombination. The authors propose that this leads to ALT, based on increased TSCs and accumulation of telomerase in APBs. The authors propose that TGS1 assures the telomerase-dependent TMM and suppressed ALT, which they suggest underpins a novel mechanism controlling the choice of TMM.

Overall, I am convinced by the data presented in this paper and concur with the conclusion that TGS1 is important for the appropriate engagement of telomerase with telomere ends to allow productive TM.

Minor points:

1. I do not believe that TGS1 "directs a novel mechanism that instructs the choice of TMM pathway". Rather, loss of TGS1 compromises telomerase-dependent TMM, and this culminates in ExoI-dependent resection of telomeres that creates a substrate for HR. The authors should tone down the "choice of TMM pathway" conclusions as this is unsubstantiated.
2. The authors claim that loss of TGS1 leads to ALT. While this is suggested by increased APBs and TSCs, I would advise that the authors change this to "features of ALT" as only a subset of ALT features have been explored.
3. The authors should examine C-circles and localisation of Rad51 foci to telomeres following TGS1 depletion.

Reviewer #2 (Remarks to the Author):

Buemi et al., have shown that TGS1 mediated modification of TERC is essential for the telomerase mediated telomere elongation process. The authors performed impressive amount of experiments to investigate this using various model systems including organoid. However, there are several points to be clarified and corrected as written below.

1. The authors performed the Interphase telomere-FISH using Cy3-labeled (CCCTAA)₃ to detect the G-rich sequences and Alexa488-labelled (TTAGGG)₃ probe to detect the C-rich sequences. In normal interphase telomere-FISH condition, the leading strand and lagging strands are not distinguishable. For example, C-rich sequences telomere FISH signal (using TTAGGG probe) represents the parental strand of leading strand and the daughter strand of lagging strand. G-rich sequence telomere FISH signal (using CCCTAA probe) represents the daughter strand of leading strand and the parental strand of lagging strand. The authors should correct their terminology: "lagging strand" should be changed to "G-rich sequences" and "leading strand" should be changed to "C-rich sequences". Because the author performed interphase telomere-FISH without removing daughter strands.
2. Recently, it is reported that TGS1 is responsible for the trimethylation capping of TERC (Chen et al., 2020 Cell reports, PMID: 32023455). They demonstrated that loss of TGS1 led to the increase in TERC level and telomerase activity consistent with the author's data in this manuscript. The increased TERC

level and enhanced telomerase activity, consistently the telomere length has been elongated shown as TRF gel.

3. In Figure 3, the elongated G-rich sequences can be derived from excessive telomere elongation by enhanced telomerase activity in TGS1 depleted or SINEFUNGIN-treated cells.

4. Because the authors have tested the effect of TGS1 depletion, SINEFUNGIN in telomerase overexpressing H1299 (presumably their telomere length very long), in Figure 4, the increased TRF2 and PML co-localization can be derived from telomere trimming events (Pickett et al., 2009 EMBO J., PMID: 19214183) due to enhanced telomerase activity in TGS1 depleted or SINEFUNGIN-treated cells.

5. It would be more informative if the author can demonstrate the TRF gel image for Figure 5L-N. Two-fold telomere intensity increase should make significant difference in TRF gel.

6. Page 3 line 51: 3'end of the telomere lagging strand: In human, telomerase mediated telomere elongation is occurring after DNA replication processes in both leading and lagging strands. "Lagging strand" should be changed as "G-rich sequences (TTAGGG)".

Reviewer #3 (Remarks to the Author):

Overview: Cellular immortality requires the activation of a Telomere Maintenance Mechanism (TMM) to counteract normal telomere shortening that occurs through each cycle of DNA replication and cell division. For approximately 85% of cancers telomere maintenance is provided by the RNA-protein enzyme complex telomerase; the remaining ~15% use the Alternative Lengthening of Telomeres (ALT) TMM, which operates through a recombination-based mechanism. As normal somatic cells generally lack a TMM, targeting TMMs is considered a promising avenue for cancer therapy.

A key outstanding question in telomere biology is the nature of the genetic and/or molecular mechanisms that dictate which TMM is activated. A related question, specifically relevant to potential therapeutic application, is whether inhibition of telomerase would drive the activation of ALT, thereby allowing continued proliferation of the cancer cells. It is this question of specific activation of one TMM over another that this work addresses.

In this work, the authors explore the functional consequences from formation of a 2,2,7-trimethylguanosine 5'-cap on the human telomerase RNA component (hTR) mediated by the methylase TGS1. Inhibiting this trimethylation, either by RNA knockdown of TGS1 or enzymatic inhibition with a small molecule, impaired telomerase recruitment to the telomere and disrupted telomere homeostasis. Most significantly, the authors conclude that the presence of the trimethyl cap on hTR is a molecular feature that suppresses the ALT mechanism, and inhibiting the trimethylation activates ALT in these cells.

Reviewer's assessment: This work contains many interesting observations with respect to the activation of the telomerase-based TMM and the role of the 2,2,7-trimethylguanosine cap of hTR. However, their primary and certainly most significant conclusion – that inhibiting formation of the trimethyl cap leads to ALT activation – is not supported by the data: simply put, the authors do not show ALT activity.

ALT cells display the following properties (see the section "Phenotypic characteristics of ALT cells" in reference #3):

- An abundance of extra-chromosomal circular telomeric DNA, notably "C-circles", which forms the basis of an established and quantitative means to detect ALT activity: Henson JD, et al. (2009) "DNA C-circles are specific and quantifiable markers of alternative-lengthening-of-telomeres activity" Nature

Biotechnology, 12: 1181-1185.

- Highly heterogeneous chromosomal telomere length, including extremely long telomeres relative to telomeres in telomerase+ cells.
- ALT-associated PML bodies (APBs): Telomeric DNA (chromosomal or extrachromosomal) are present in a subset of promyelocytic leukaemia nuclear bodies (PML bodies), indicated by the colocalization of telomeric DNA and the protein PML in a spherical nuclear body.
- ALT cells synthesise telomeric DNA in APBs outside of S-phase (G2; sometimes referred to as the "ATSA assay"): Zhang J et al. (2019) "Alternative lengthening of telomeres through two distinct break-induced replication pathways" Cell Reports, 26: 955-968.

The authors do not perform C-circle assays to support the presence of ALT activity. They observe a modest (~10-15%) increase in telomere length (Figure 3), but this is hardly indicative of the dramatic differences in telomere length typically observed between telomerase+ and ALT cells.

They also suggest an increase in the abundance of APBs upon TGS1 inhibition (Figure 4). First of all, APBs are very rarely observed in unperturbed telomerase+ cells, so the presence of a substantial background of APBs in their "control" telomerase+ cells is curious. From what I can tell by the Methods the authors do not appear to be performing Z-stacking in their IF???. Second, the increase in APBs upon TGS1 inhibition, while "statistically significant" (~2-fold over control), appears modest for a genuine ALT cell. Third, perturbing telomerase+ cells in a manner that increases telomere length has been shown to induce formation of APBs. The authors should consult: Pickett HA, et al. (2009) "Control of telomere length by a trimming mechanism that involves generation of t-circles" EMBO J. 28: 799-809. Specifically, Figure 3 illustrates proper confirmation of APBs by deconvolution of Z-stacked images and 3D imaging. Fourth, Supp Figure 5A shows that total PML bodies per nucleus increases upon TGS1 inhibition, and this increase mirrors the "increase in APBs"; could the "increase in APBs" be attributed simply to statistical chance of having more PML bodies?

The authors also suggest an impairment of telomere recruitment pathways, which transit through Cajal bodies, upon TGS1 inhibition by measuring the colocalization of the protein coilin (marker of Cajal bodies) and TRF2 (marker for telomeres). A "decrease in Cajal-TRF2 colocalizations" is observed upon TGS1 inhibition. But analogous to my fourth point above, the total number of Cajal bodies per nucleus also decreases upon TGS1 inhibition, and this decrease mirrors the change in coilin-TRF2 colocalizations, rendering their conclusion equivocal.

My final concern is at a conceptual level: As the authors correctly note in reference #18, genetic inhibition of telomerase activity in a telomerase+ cell line resulted in the emergence of clones that had activated ALT to support continued proliferation. However, this was an extremely rare event – three clones from 12 million cells. Furthermore, populations of 10^8 cells of two other telomerase+ cell lines failed to produce any surviving ALT clones. So I find it unlikely that modest changes in telomerase recruitment and/or telomere length dynamics (without actually inhibiting telomerase activity) could result in activation of ALT.

A suggestion for the authors regarding telomere length measurements: Most researchers in the ALT field show telomere length by the southern blotting technique of Telomere Restriction Fragment, providing a visual presentation of telomere length distributions. But even with the present data in hand, I would suggest using a linear, rather than logarithmic, y-axis to better illustrate telomere length changes; likewise, violin plots may also convey distributions better than box-whisker plots.

Scott Cohen, PhD

Buemi et al. 2021

DETAILED ANSWERS TO REVIEWERS' COMMENTS

Reviewer #1 (Remarks to the Author):

In this study, Buemi and co-workers investigate the role of Trimethylguanosine Synthetase 1 (TGS1) in telomerase-dependent telomere maintenance mechanisms (TMM) in human cells. Through a series of well-designed and present experiments, the authors convincingly demonstrate that TGS1 is important for TMM in human cells. They show that TGS1 facilitates the formation of a 2,2,7-TMG cap at the 5' end of telomerase hTR non-coding RNA, which directs the telomerase holoenzyme to Cajal bodies and chromosome ends to allow product TMM. In support of this conclusion, depletion of TGS1 or inhibition by Sinefungin blocks telomerase recruitment to telomeres but does not impact of enzymatic activity per se. In the absence/inhibition of TGS1, telomere ends are subject to Exo1 dependent resection, resulting in increased 3' single stranded DNA, which engaged in Rad51 dependent homologous recombination. The authors propose that this leads to ALT, based on increased TSCs and accumulation of telomerase in APBs. The authors propose that TGS1 assures the telomerase-dependent TMM and suppressed ALT, which they suggest underpins a novel mechanism controlling the choice of TMM.

Overall, I am convinced by the data presented in this paper and concur with the conclusion that TGS1 is important for the appropriate engagement of telomerase with telomere ends to allow productive TMM.

ANSWER TO REVIEWER:

We thank reviewer 1 for classifying our manuscript as important work on the understanding of the engagement of telomerase with chromosome ends.

Minor points:

REVIEWER COMMENT #1: I do not believe that TGS1 “directs a novel mechanism that instructs the choice of TMM pathway”. Rather, loss of TGS1 compromises telomerase-dependent TMM, and this culminates in Exo1-dependent resection of telomeres that creates a substrate for HR. The authors should tone down the “choice of TMM pathway” conclusions as this is unsubstantiated.

ANSWER TO REVIEWER COMMENT #1: The revised version of the manuscript contains new data on ALT features, triggered by loss of TGS1. In particular, we found that loss of TGS1 leads to the formation of DNA:RNA hybrids, C-circles, recruitment of the BLM helicase and activation of ATR and also promotes the recruitment of POLD3, is an accessory subunit of the replicative Pol δ polymerase (for details, please see answer to reviewer comment #2).

We agree with the reviewer to tone down our statement of TMM choice. We state now:

- in the abstract:

“This indicates a critical role for 2,2,7-TMG capping of the RNA component of human telomerase (hTR) in enforcing telomerase dependent telomere maintenance to suppress ALT.” (Page 2, line 11),

- in the last paragraph of the main text:

“Our data highlight a central role for TGS1 dependent 2,2,7-TMG capping of hTR to assure selective, telomerase-dependent telomere maintenance and suppress aberrant activation of the ALT pathway in telomerase positive cancer cells.”. (Page 10, line 20).

REVIEWER COMMENT #2. The authors claim that loss of TGS1 leads to ALT. While this is suggested by increased APBs and TSCs, I would advise that the authors change this to “features of ALT” as only a subset of ALT features have been explored.

ANSWER TO REVIEWER COMMENT #2. In the revised version of the manuscript we show that loss of TGS1 function mediates the activation of additional, classic features of the ALT pathway in telomerase positive H1299 cells.

In particular we show in:

- **NEW FIGURE 5A-C:** Loss of TGS1 function leads to increased DNA:RNA hybrid levels at telomeres (PAGE 7, Line 27)
- **NEW FIGURE 5D, E and NEW SUPPLEMENTARY FIG. 6A, B:** Loss of TGS1 function leads to increased ATR activation at telomeres of experimental H1299 cells and nuclear P-ATR in sinefungin-treated tumor-organoids. (PAGE 7, Line 31)
- **NEW FIGURE 5L, M and NEW SUPPLEMENTARY FIGURE 7G, H:** Loss of TGS1 function promotes C-circle formation in telomerase positive H1299 cells. (PAGE 8, Line 13); (PAGE 9, Line 14)
- **NEW FIGURE 5F-G:** Loss of TGS1 function promotes the recruitment of RAD51 to telomeres. (PAGE 8, Line 1)
- **NEW FIGURE 5J-K and NEW SUPPLEMENTARY FIG. 6C, SUPPLEMENTARY FIG. 7D-F:** Loss of TGS1 function recruits BLM to telomeres, a feature known to promote ALT. (PAGE 8, Line 4); (PAGE 9, Line 13)
- **NEW FIGURE 6O, P:** Loss of TGS1 function promotes POLD3 recruitment to telomeres in 5--Aza-2'deoxyctidine treated cells. (PAGE 9, Line 25)

This new body of evidence supports a role for TGS1 in suppressing ALT in telomerase positive cancer cells.

REVIEWER COMMENT #3. The authors should examine C-circles and localisation of Rad51 foci to telomeres following TGS1 depletion.

ANSWER TO REVIEWER COMMENT #3. The requested dataset was added to the final version of the manuscript

- RAD51 colocalization with telomeres is shown in **NEW FIGURE 5F, G.** Corresponding data is explained in the main text (PAGE 8, Line 1).
- C-circles in TGS1 loss of function conditions can be found in **NEW FIGURE 5L, M and NEW SUPPLEMENTARY FIGURE 7G, H.** (PAGE 8, Line 13) (PAGE 9, Line 14).

We found that loss of TGS1 mediates increased C-circle levels and promotes RAD51 localization at telomeres.

Reviewer #2 (Remarks to the Author):

Buemi et al., have shown that TGS1 mediated modification of TERC is essential for the telomerase mediated telomere elongation process. The authors performed impressive amount of experiments to investigate this using various model systems including organoid. However, there are several points to be clarified and corrected as written below.

ANSWER TO REVIEWER:

We thank reviewer 2 for stating that our manuscript contains a large amount of data including patient derived precision cancer medicine model systems and providing extremely useful suggestions for the improvement of our manuscript.

REVIEWER COMMENT #1. The authors performed the Interphase telomere-FISH using Cy3-labeled (CCCTAA)₃ to detect the G-rich sequences and Alexa488-labelled (TTAGGG)₃ probe to detect the C-rich sequences. In normal interphase telomere-FISH condition, the leading strand and lagging strands are not distinguishable. For example, C-rich sequences telomere FISH signal (using TTAGGG probe) represents the parental strand of leading strand and the daughter strand of lagging strand. G-rich sequence telomere FISH signal (using CCCTAA probe) represents the daughter strand of leading strand and the parental strand of lagging strand. The authors should correct their terminology: “lagging strand” should be changed to “G-rich sequences” and “leading strand” should be changed to “C-rich sequences”. Because the author performed interphase telomere-FISH without removing daughter strands.

ANSWER TO REVIEWER COMMENT #1:

We have introduced the requested change throughout the manuscript.

REVIEWER COMMENT #2. Recently, it is reported that TGS1 is responsible for the trimethylation capping of TERC (Chen et al., 2020 Cell reports, PMID: 32023455). They demonstrated that loss of TGS1 led to the increase in TERC level and telomerase activity consistent with the author’s data in this manuscript. The increased TERC level and enhanced telomerase activity, consistently the telomere length has been elongated shown as TRF gel.

ANSWER TO REVIEWER COMMENT #2:

The respective publication has been cited in our manuscript (REF. 28). In Chen et al. TGS1 has been deleted by CRISPR/Cas9 technology in UMUC3 cells. A telomere length-increase in cell clones obtained from CRISPR modified single cells (with a very long replication history) was demonstrated by denaturing TRF using a radiolabeled CCCTAA probe that hybridizes to the G-rich telomere strand. No information on the length homeostasis of the C-rich telomere strand is shown. It was anticipated that increased telomerase activity present in TGS1 loss of function cells mediates alterations in telomere length, however this was not experimental validated; further, ALT features were not addressed in the respective cell model.

In our manuscript we were interested in understanding events occurring in a short time window right after TGS1 loss of function. In our experimental time windows (max. 10 days) we can detect alterations in telomere length by single telomere strand specific Q-FISH which is more sensitive than TRF. Maximum alterations in telomere length alterations in TGS1 loss of function experiments were +15% for the G-rich strand and -40% for the C-rich strand. The resulting expanded overhang of the G-rich

telomere demonstrated by a native TRF following a modified protocol from the Shay group (PMID: 28082393): **NEW SUPPLEMENTARY FIGURE 4C, D (PAGE 6, Line 16)**

We performed classic, denaturing TRFs using telomere probes, however we found that – not having access to pulsed field electrophoresis equipment - the method was not sufficiently sensitive to provide a clear information on telomere length alterations.

To provide additional evidence to support Q-FISH data shown in the original version of the manuscript, we performed additional, new Q-FISH analyses of H1299 cells and Clone 22 cells shown in the **NEW SUPPLEMENTARY FIGURE 4E-L**. Notably, these experiments also show that telomere alterations are not dependent on telomerase (hTERT). (**PAGE 6, Line 21**)

We discuss the implication of our data on the previously published study by Chen et al. on **PAGE 10, Line 13** of the revised version of our manuscript.

REVIEWER COMMENT #3. In Figure 3, the elongated G-rich sequences can be derived from excessive telomere elongation by enhanced telomerase activity in TGS1 depleted or SINEFUNGIN-treated cells.

ANSWER TO REVIEWER COMMENT #3: In order to address this issue, we have knocked-down hTERT and TGS1 expression on parental H1299 cells and knocked down hTERT in H1299 clone 22 cells (overexpressing hTR and hTERT) that were treated with sinefungin or left untreated. We found that acute depletion of hTERT mimicked the defect of recruitment of hTERT to telomeres observed in TGS1 loss of function cells.

Importantly, telomere single strand alterations in TGS1 loss of function models remained unaltered when hTERT was knocked down. We conclude that, telomerase does not impact on telomere single strand alterations in TGS1 loss of function cells. This data is shown as **NEW SUPPLEMENTARY FIGURE 4E-L** and described on **PAGE 6, Line 21**.

REVIEWER COMMENT #4. Because the authors have tested the effect of TGS1 depletion, SINEFUNGIN in telomerase overexpressing H1299 (presumably their telomere length very long), in Figure 4, the increased TRF2 and PML co-localization can be derived from telomere trimming events (Pickett et al., 2009 EMBO J., PMID: 19214183) due to enhanced telomerase activity in TGS1 depleted or SINEFUNGIN-treated cells.

ANSWER TO REVIEWER COMMENT #4:

In the mentioned study of the Reddel group, ectopic expression of hTR resulted a stable, doubling of telomere length in telomerase positive cancer cells. Authors describe increased APB numbers, ECTR DNA, absence of T-SCE and telomere-telomere copying and a lack of DNA damage at telomeres. The existence of a mechanism was proposed that limits uncontrolled telomere elongation by a trimming mechanism that involves ECTR DNA formation. Possible off-target effects of hTR expression or telomerase were not addressed.

For the following reasons we feel that our data does not support a model where telomere length issues may interfere with phenotypes resulting from TGS1 loss of function experiments:

1. Loss of TGS1 induces a shortening of the C-rich strand and modest, 10-15% increase of the G-rich (Fig. 3A-D, Fig. 3H, I; Fig 6A-D; **NEW SUPPLEMENTARY FIGURE 4E-L**). Although we do not observe an elongation of both telomere strands, we can observe multiple features of the ALT pathway in TGS1 loss of function cells (see below, 2-6).
2. Loss of TGS1 leads to elevated T-SCE (Fig. 4I, J), which was not observed in the Pickett et al study.

3. We found that loss of TGS1 leads to DNA:RNA hybrid formation and activation of the DNA damage signaling kinase ATR at telomeres (**NEW FIGURE 5A-C**). The Pickett et al study did not observe DNA damage signaling at long telomeres.
4. In addition to this, we observed the recruitment of several other classic marks found at ALT telomeres, including BLM and RAD51 recruitment (**NEW FIGURE 5F, G, J, K**).
5. Elongation of both telomere single strands in TGS1 loss of function cells was obtained only when cells were treated also with 5-Aza-2'deoxyctidine to stimulate APB formation. In this context telomeres showed improved recruitment to PML bodies, colocalization with BLM and recruitment of POLD3 providing a ratio for telomere elongation (**NEW FIGURE 6 O, P; NEW FIGURE 5 H, I; NEW SUPPLEMENTARY FIGURE 7D-F**).
6. We further excluded a possible role of telomerase (hTERT) in modulating the localization of telomeres to PML bodies or telomere length alterations in TGS1 loss of function cells (**NEW SUPPLEMENTARY FIGURE 4E-L; NEW SUPPLEMENTARY FIGURE 5D-F**)

Altogether, these findings support that observed telomere phenotypes are a direct consequence of TGS1 loss of function.

REVIEWER COMMENT #5. It would be more informative if the author can demonstrate the TRF gel image for Figure 5L-N. Two-fold telomere intensity increase should make significant difference in TRF gel.

ANSWER TO REVIEWER COMMENT #5:

We did detect a 35%-40% increase of length of both telomere strands in cells treated with sinefungin and 5'Aza-2'deoxyctidine for 10 days; not a 2 fold increase, as state by the reviewer.

We have performed denaturing and native TRF analysis. Native TRF confirmed an extension of G-strand overhangs in TGS1 loss of function experiments (**NEW SUPPLEMENTARY FIGURE 4C, D**). We were performing denaturing TRFs aiming to demonstrate a 35% - 40% increase in telomere length observed in cells treated with sinefungin and 5'Aza-2'deoxyctidine for 10 days. However, we found that the method was not sufficiently sensitive to provide a clear information on telomere length alterations (unfortunately, we do not have access to pulsed field gel electrophoresis).

However, we want to highlight that telomere length alterations were recapitulated in additional, new Q-FISH analyses of H1299 cells and Clone 22 cells shown in the **NEW SUPPLEMENTARY FIGURE 4E-L**. Given that Q-FISH is a standard technique for telomere length measurements we are convinced to provide correct information of telomere single strand alterations in our TGS1 loss of function models.

REVIEWER COMMENT #6. Page 3 line 51: 3'end of the telomere lagging strand: In human, telomerase mediated telomere elongation is occurring after DNA replication processes in both leading and lagging strands. "Lagging strand" should be changed as "G-rich sequences (TTAGGG)".

ANSWER TO REVIEWER COMMENT #6:

We have introduced the requested change throughout the manuscript.

Reviewer #3 (Remarks to the Author):

Overview: Cellular immortality requires the activation of a Telomere Maintenance Mechanism (TMM) to counteract normal telomere shortening that occurs through each cycle of DNA replication and cell division. For approximately 85% of cancers telomere maintenance is provided by the RNA-protein enzyme complex telomerase; the remaining ~15% use the Alternative Lengthening of Telomeres (ALT) TMM, which operates through a recombination-based mechanism. As normal somatic cells generally lack a TMM, targeting TMMs is considered a promising avenue for cancer therapy.

A key outstanding question in telomere biology is the nature of the genetic and/or molecular mechanisms that dictate which TMM is activated. A related question, specifically relevant to potential therapeutic application, is whether inhibition of telomerase would drive the activation of ALT, thereby allowing continued proliferation of the cancer cells. It is this question of specific activation of one TMM over another that this work addresses.

In this work, the authors explore the functional consequences from formation of a 2,2,7--trimethylguanosine 5'-cap on the human telomerase RNA component (hTR) mediated by the methylase TGS1. Inhibiting this trimethylation, either by RNA knockdown of TGS1 or enzymatic inhibition with a small molecule, impaired telomerase recruitment to the telomere and disrupted telomere homeostasis. Most significantly, the authors conclude that the presence of the trimethyl cap on hTR is a molecular feature that suppresses the ALT mechanism, and inhibiting the trimethylation activates ALT in these cells.

ANSWER TO REVIEWER:

We thank the reviewer for elegantly putting our manuscript in the context of latest progress of research on telomere maintenance mechanisms. Reviewer 3 raises important issues with regards to the activation of hallmark features of ALT in our cell model systems. In the revised version of our manuscript we provide now multiple lines of evidence that show that TGS1 is important to suppress ALT in telomerase positive cancer cells.

REVIEWER COMMENT #1

Reviewer's assessment: This work contains many interesting observations with respect to the activation of the telomerase-based TMM and the role of the 2,2,7-trimethylguanosine cap of hTR. However, their primary and certainly most significant conclusion – that inhibiting formation of the trimethyl cap leads to ALT activation – is not supported by the data: simply put, the authors do not show ALT activity. ALT cells display the following properties (see the section “Phenotypic characteristics of ALT cells” in reference #3):

- An abundance of extra-chromosomal circular telomeric DNA, notably “C-circles”, which forms the basis of an established and quantitative means to detect ALT activity: Henson JD, et al. (2009) “DNA C-circles are specific and quantifiable markers of alternative-lengthening-of-telomeres activity” *Nature Biotechnology*, 12: 1181-1185.

ANSWER TO REVIEWER COMMENT: C-circle analysis on TGS1 loss of function cells are shown in **NEW FIGURE 5L, M** and **NEW SUPPLEMENTARY FIGURE 7G, H** of the revised version of our manuscript.

The classic ALT cell line U-2 OS was used as positive control in the experiment and displays very high C-circle levels when compared to our H1299 model cells that display only almost undetectable basal C-

circle levels. This is presumably due to the optimized ALT setup on U-2 OS cells when compared to telomerase positive H1299 cells.

Importantly, TGS1 loss of function results in significant increase in C-circle abundance in TGS1 loss of function H1299 cells. The detection of C-circles is in line with the emergence of additional ALT markers in TGS1 loss of function cells (please see below). The respective text can be found on **PAGE 8, Line 13** and **PAGE 9, Line 14**).

- Highly heterogeneous chromosomal telomere length, including extremely long telomeres relative to telomeres in telomerase+ cells.

ANSWER TO REVIEWER COMMENT: Telomere Q-FISH analysis revealed a change in telomere length distribution when cells were treated with sinesfungin and 5'Aza-2'deoxyctidine (both, C-rich and G-rich strands). Our data show a particular increase in the fraction of longest telomeres, thus supporting observations in ALT cells (see **NEW SUPPLEMENTARY FIGURE 7 I, J**). The respective text can be found on **PAGE 9, Line 18**.

- ALT-associated PML bodies (APBs): Telomeric DNA (chromosomal or extrachromosomal) are present in a subset of promyelocytic leukaemia nuclear bodies (PML bodies), indicated by the colocalization of telomeric DNA and the protein PML in a spherical nuclear body.

ANSWER TO REVIEWER COMMENT: We have included colocalization data on telomeres and PML in the revised version of our manuscript. This dataset, obtained by confocal microscopy is shown in **NEW SUPPLEMENTARY FIG 5D-F**. We confirm by immunoFISH that loss of TGS1 function results in increased PML-telomere colocalization frequencies. The respective text can be found on **PAGE 7, Line 13**.

- ALT cells synthesise telomeric DNA in APBs outside of S-phase (G2; sometimes referred to as the "ATSA assay"): Zhang J et al. (2019) "Alternative lengthening of telomeres through two distinct break-induced replication pathways" Cell Reports, 26: 955-968.

ANSWER TO REVIEWER COMMENT: We have not addressed DNA synthesis in APBs in experimental model cell. However, we show in **NEW FIGURE 6O, P** that loss of TGS1 function and treatments with 5-Aza-2'deoxyctidine promotes POLD3 recruitment to telomeres. This provides a rational for the elongation of both C-rich and G-rich telomere strands in this context. (**PAGE 9, Line 25**).

In addition to this, we provide additional evidence that TGS1 suppresses the ALT pathway:

- **NEW FIGURE 5A-C**: Loss of TGS1 function leads to increased DNA:RNA hybrid levels at telomeres. (**PAGE 7, Line 27**)
- **NEW FIGURE 5L, M** and **NEW SUPPLEMENTARY FIGURE 7G, H**: Loss of TGS1 function promotes C-circle formation. (**PAGE 8, Line 13**); (**PAGE 9, Line 14**)
- **NEW FIGURE 5J-K** and **Supplementary Fig. 6C, Supplementary fig. 7D-F**: Loss of TGS1 function recruits BLM to telomeres, a feature known to promote ALT. (**PAGE 8, Line 4**); (**PAGE 9, Line 13**)
- **NEW FIGURE 5D, E** and **Supplementary fig. 6A, B**: Loss of TGS1 function leads to increased ATR activation in tumor organoids and increased ATR activation at telomeres of experimental H1299 cells. (**PAGE 7, Line 31**)
- **NEW FIGURE 5F-G**: Loss of TGS1 function promotes the recruitment of RAD51 to telomeres. (**PAGE 8, Line 1**)

REVIEWER COMMENT #2

The authors do not perform C-circle assays to support the presence of ALT activity. They observe a modest (~10-15%) increase in telomere length (Figure 3), but this is hardly indicative of the dramatic differences in telomere length typically observed between telomerase+ and ALT cells.

ANSWER TO REVIEWER COMMENT: #2

C-circle analysis on TGS1 loss of function cells are shown in **NEW FIGURE 5L, M** and **NEW SUPPLEMENTARY FIGURE 7G, H** of the revised version of our manuscript. The classic ALT cell line U-2 OS was used as positive control in the experiment and displays very high C-circle levels when compared to our H1299 model cells that display only almost undetectable basal C-circle levels. This is presumably due to the optimized ALT setup on U-2 OS cells when compared to telomerase positive H1299 cells. Importantly, TGS1 loss of function results in significant increase in C-circle abundance in TGS1 loss of function H1299 cells. The detection of C-circles is in line with the emergence of additional ALT markers in TGS1 loss of function cells (please see below, Answer to reviewer comment #4). The respective text can be found on **PAGE 8, Line 13** and **PAGE 9, Line 14**.

REVIEWER COMMENT #3: They also suggest an increase in the abundance of APBs upon TGS1 inhibition (Figure 4). APBs are very rarely observed in unperturbed telomerase+ cells, so the presence of a substantial background of APBs in their “control” telomerase+ cells is curious. From what I can tell by the Methods the authors do not appear to be performing Z-stacking in their IF???

ANSWER TO REVIEWER COMMENT: We have included new colocalization data on PML and telomeres obtained by immuno-FISH and confocal microscopy (**NEW SUPPLEMENTARY FIGURE 5, D-F**). These experiments also contain additional hTERT knock down experiments and confirm our initial data. In particular, we were able to detect a PML body in 50% of untreated cells; of those PML bodies only 5% appear to colocalize with TRF2; TGS1 loss of function increases PML body numbers but also the frequency (in %) of PML bodies colocalizing with telomeres. Together data from Fig. 4A-D and **NEW SUPPLEMENTARY FIGURE 5, D-F** are indicative for the promotion APB formation in TGS1 loss of function cells. The respective text can be found on **PAGE 7, Line 12**.

REVIEWER COMMENT #4: The increase in APBs upon TGS1 inhibition, while “statistically significant” (~2-fold over control), appears modest for a genuine ALT cell.

ANSWER TO REVIEWER COMMENT #4: We agree that the number of APBs in H1299 cells with impaired TGS1 function are lower when compared to classic ALT cells, such as U-2 OS cells. U-2 OS cells show an “optimized” ALT setup that is due to specific mutations (such as DAXX/ATRAX), high TERRA expression and abnormal chromatin structure that promote homologous recombination in the absence of telomerase. Our experimental focusses on cancer cell models with strong telomerase-dependent TMM that use several pathways that function in a redundant manner to block ALT and formation of APBs (DNA methylation, low TERRA expression, ATRX/DAXX wildtype,...).

This may also be the reason why TGS1 loss of function H1299 cells does not manifest APB numbers found in U-2 OS cells. In the revised version of our manuscript we present additional data that support an activation of ALT pathway in TGS1 loss of function cells:

- **NEW FIGURE 5A-C**: Loss of TGS1 function leads to increased DNA:RNA hybrid levels at telomeres. (**PAGE 7, Line 27**)

- **NEW FIGURE 5L, M and NEW SUPPLEMENTARY FIGURE 7G, H:** Loss of TGS1 function promotes C-circle formation. (PAGE 8, Line 13); (PAGE 9, Line 14)
- **NEW FIGURE 5J-K and Supplementary Fig. 6C, Supplementary fig. 7D-F:** Loss of TGS1 function recruits BLM to telomeres, a feature known to promote ALT. (PAGE 8, Line 4); (PAGE 9, Line 13)
- **NEW FIGURE 5D, E and Supplementary fig. 6A, B:** Loss of TGS1 function leads to increased ATR activation in tumor organoids and increased ATR activation at telomeres of experimental H1299 cells. (PAGE 7, Line 31)
- **NEW FIGURE 5F-G:** Loss of TGS1 function promotes the recruitment of RAD51 to telomeres. (PAGE 8, Line 1)
- **NEW FIGURE 6O, P:** Loss of TGS1 function promotes POLD3 recruitment to telomeres in 5-Aza-2'-deoxycytidine treated cells. This provides a rationale for the elongation of both C-rich and G-rich telomere strands in this context (PAGE 9, Line 25).

REVIEWER COMMENT #5: Perturbing telomerase+ cells in a manner that increases telomere length has been shown to induce formation of APBs. The authors should consult: Pickett HA, et al. (2009) "Control of telomere length by a trimming mechanism that involves generation of t-circles" EMBO J. 28: 799-809. Specifically, Figure 3 illustrates proper confirmation of APBs by deconvolution of Z-stacked images and 3D imaging.

ANSWER TO REVIEWER COMMENT #5:

In the mentioned study by the Reddel group, ectopic expression of hTR resulted a stable, doubling of telomere length in telomerase positive cancer cells. Authors describe increased APB numbers, ECTR DNA, absence of T-SCE, telomere-telomere copying and a lack of DNA damage at telomeres. The existence of a mechanism was proposed that limits uncontrolled telomere elongation by a trimming mechanism that involves ECTR DNA formation. Possible off-target effects of hTR expression or telomerase were not addressed.

For the following reasons we feel that our data does not support a model where telomere length issues may interfere with phenotypes resulting from TGS1 loss of function experiments:

1. Loss of TGS1 induces a shortening of the C-rich strand and modest, 10-15% increase of the G-rich (Fig. 3A-D, Fig. 3H, I; Fig 6A-D; **NEW SUPPLEMENTARY FIGURE 4E-L**). Although we not observe an elongation of both telomere strands, we can observe multiple features of the ALT pathway in TGS1 loss of function cells (see below, 2-6).
2. Loss of TGS1 leads to elevated T-SCE (Fig. 4I, J), which was not observed in the Pickett et al study.
3. We found that loss of TGS1 leads to DNA:RNA hybrid formation and activation of the DNA damage signaling kinase ATR at telomeres (**NEW FIGURE 5D-E**). The Pickett et al study did not observe DNA damage signaling at long telomeres.
4. In addition to this we observed the recruitment of several other classic marks found at ALT telomeres, including BLM and RAD51 recruitment (**NEW FIGURE 5F, G, J, K**).
5. Elongation of both telomere single strands in TGS1 loss of function cells was obtained only when cells were treated also with 5-Aza-2'-deoxycytidine to stimulate APB formation. In this context telomeres showed improved recruitment to PML bodies, co-localization with BLM and recruitment of POLD3 providing a ratio for telomere elongation (**NEW FIGURE 6O, P; NEW FIGURE 5J, K; NEW SUPPLEMENTARY FIGURE 7D-F**).
6. PML-telomere colocalization was confirmed by telomere immunoFISH (analyzed by confocal microscopy): **NEW SUPPLEMENTARY FIGURE 5D-F**.

7. We further excluded a possible role of telomerase (hTERT) in modulating the localization of telomeres to PML bodies or telomere length alterations in TGS1 loss of function cells (**NEW SUPPLEMENTARY FIGURE 4E-L; NEW SUPPLEMENTARY FIGURE 5D-F**)

Altogether, these findings support that observed telomere phenotypes are a direct consequence of TGS1 loss of function.

REVIEWER COMMENT #6: Supp Figure 5A shows that total PML bodies per nucleus increases upon TGS1 inhibition, and this increase mirrors the “increase in APBs”; could the “increase in APBs” be attributed simply to statistical chance of having more PML bodies?

ANSWER TO REVIEWER COMMENT #6:

In our experiment the numbers of PML bodies per cell nucleus as well as % of PML bodies that colocalize with telomeres (telomere repeats and TRF2) are shown (Figure 4A-D, **NEW SUPPLEMENTARY FIGURE 5D-F**). We were able to detect a PML body in 50% of untreated cells; of those only 5% appear to colocalize with TRF2. This would be consistent with the absence of ALT in telomerase positive H1299 cells. Importantly, TGS1 loss of function increases PML body numbers but at the same time also augments the frequency (in %) of PML bodies colocalizing with telomeres. This indicates increased rates of engagement of PML bodies with telomeres.

REVIEWER COMMENT #7: The authors also suggest an impairment of telomere recruitment pathways, which transit through Cajal bodies, upon TGS1 inhibition by measuring the colocalization of the protein coilin (marker of Cajal bodies) and TRF2 (marker for telomeres). A “decrease in Cajal-TRF2 colocalizations” is observed upon TGS1 inhibition. But analogous to my fourth point above, the total number of Cajal bodies per nucleus also decreases upon TGS1 inhibition, and this decrease mirrors the change in coilin-TRF2 colocalizations, rendering their conclusion equivocal.

ANSWER TO REVIEWER COMMENT #7:

In our experiment the numbers of Cajal bodies per cell nucleus as well as % of Cajal bodies that colocalize with telomeres are shown Figure 2A-H. We were able to detect ca. 2 Cajal body per nucleus in untreated cells; of those 15% appear to colocalize with TRF2. This is consistent with telomerase dependent TMM in H1299 cells. Importantly, TGS1 loss of function decreases Cajal body numbers but at the same time also reduce the frequency (in %) of telomeres co-localizing with Cajal bodies. Thus, the reduced rates of engagement of Cajal bodies with telomeres supports a suppression of telomerase depended TMM in our model cells.

REVIEWER COMMENT #8: My final concern is at a conceptual level: As the authors correctly note in reference #18, genetic inhibition of telomerase activity in a telomerase+ cell line resulted in the emergence of clones that had activated ALT to support continued proliferation. However, this was an extremely rare event – three clones from 12 million cells. Furthermore, populations of 10^8 cells of two other telomerase+ cell lines failed to produce any surviving ALT clones. So I find it unlikely that modest changes in telomerase recruitment and/or telomere length dynamics (without actually inhibiting telomerase activity) could result in activation of ALT.

ANSWER TO REVIEWER COMMENT #8:

In the mentioned study, Min et al. used H1299 cells (also used in our study) to target hTR by CRSIPR/Cas9 technology. Targeted cells enter into crisis after 40PD and only 1 surviving clone was obtained. Thus, loss of telomerase activity impairs the viability of the vast bulk of H1299 cells by promoting critical telomere shortening in rapidly proliferating cells. The fact that only one clone was viable suggests that this clone may have acquired additional mutations that help the survival of cells by de-repressing ALT.

In our study, we focus on events that result immediately after TGS1 loss of function (in a time window of 10 days). This setup has the advantage to investigate factors that regulate the equilibrium between telomerase dependent and independent TMM in a large population of cells without imposing a dramatic selective pressure on cell survival or activation of eventual compensatory mechanisms.

Our data show that loss of TGS1 function has remarkable effects on telomere maintenance mechanisms that range from reduced telomerase recruitment, DNA:RNA hybrid formation, C-circle and ABP production, RAD51 recruitment, BLM recruitment, POLD3 recruitment to telomere sister chromatid exchange (see above). Thus, TGS1 has a novel and highly relevant function in suppressing the initiation of the ALT pathway.

In telomerase positive cancer cells multiple regulatory layers such as histone chaperons, TERRA expression, replication stress or DNA methylation ensure the suppression of ALT, thus providing an explanation the lower penetrance of ALT features in TGS1 loss of function H1299 cells when compared to a classic ALT cell line, such as U-2 OS. Enhancing ALT penetrance in H1299 cells by combining 5-Aza-2'-deoxycytidine (loss of DNA methylation) with sinefungin treatment cells support this model (Figure 6E-P, **NEW SUPPLEMENTARY FIGURE 7D-J**).

REVIEWER COMMENT #9: A suggestion for the authors regarding telomere length measurements: Most researchers in the ALT field show telomere length by the southern blotting technique of Telomere Restriction Fragment, providing a visual presentation of telomere length distributions. But even with the present data in hand, I would suggest using a linear, rather than logarithmic, y-axis to better illustrate telomere length changes; likewise, violin plots may also convey distributions better than box-whisker plots.

ANSWER TO REVIEWER COMMENT #9:

We followed the reviewer's suggestion and show now all Q-FISH data as violin plots.

After comparing data presentation as linear or logarithmic violin plots we have decided to choose logarithmic violin plots. Logarithmic violin plots allow a better visualization of the short telomere fraction (see modified Figures 3A-D; Fig 3H, I; Fig. 6A-D; **New Supplementary figure 4E-L**)

REVIEWERS' COMMENTS

Reviewer #1 (Remarks to the Author):

The authors have provided additional data to substantiate their original conclusions. I am satisfied that they have address my concerns.

Reviewer #2 (Remarks to the Author):

In this revised manuscript, the authors have addressed previous concerns by editing the text accordingly, and by performing additional experiments. Finally, there are some suggestions in the main text.

1. Page 2 line 38: homologous recombination can be changed to homology directed recombination, because homology-directed recombination includes both homologous recombination and break-induced replication. In current model, it is still unclear whether it is mediated solely by homologous recombination.

2. Page 3 line 59: breakage induced DNA replication can be changed to break-induced replication (BIR)

Reviewer #3 (Remarks to the Author):

The authors have made significant effort to improve the manuscript with the addition of several experiments requested by Reviewers. I have just a few comments:

TEXT: The role of hTR capping in promoting the telomerase TMM is well supported. The authors also associate hTR capping with "suppression" of ALT. Forgive me if this is splitting hairs, but to me the word "suppression" implies an active mechanism, which is not necessarily supported by the data. Rather, it may be that – in the absence of hTR capping – telomerase TMM becomes disfavored allowing "characteristics of ALT" or "telomeric substrates conducive to ALT" to become favored. The authors should consider rewording the text to reflect this.

My personal preference is to refrain from subjective descriptors such as "Importantly", "Remarkably", etc.

FIGURES: I appreciate the effort by the authors to pursue further lines of evidence that demonstrate features of ALT. But in Figure 5A, the high abundance of DNA:RNA hybrids (green) make discerning distinct colocalizations with TRF1 unconvincing. And in Figure 5C, the signal/noise is so low as to make this data unconvincing as well.

In Figure 5L/M (C-circle assay), the data for U2OS should also be plotted in 'M' to indicate the relative magnitude of change upon TGS1 knockdown.

Scott B Cohen, PhD

Buemi et al. 2021 - NCOMMS-20-44849A

DETAILED ANSWERS TO REVIEWERS' COMMENTS

Reviewer #1 (Remarks to the Author):

The authors have provided additional data to substantiate their original conclusions. I am satisfied that they have address my concerns.

ANSWER TO REVIEWER: We thank the reviewer for checking the manuscript and accepting the revised version of our study.

Reviewer #2 (Remarks to the Author):

In this revised manuscript, the authors have addressed previous concerns by editing the text accordingly, and by performing additional experiments. Finally, there are some suggestions in the main text.

ANSWER TO REVIEWER: We thank the reviewer for critically reading the revised version of our manuscript and high lightening last issues to be solved.

COMMENT 1: Page 2 line 38: homologous recombination can be changed to homology directed recombination, because homology-directed recombination includes both homologous recombination and break-induced replication. In current model, it is still unclear whether it is mediated solely by homologous recombination.

ANSWER TO REVIEWER: We have introduced the suggested modification. The changes are indicated in yellow and can be found on page 2, line 38-39 of the revised version of our manuscript.

COMMENT 2: Page 3 line 59: breakage induced DNA replication can be changed to break-induced replication (BIR)

ANSWER TO REVIEWER: We have introduced the suggested modification. The change is indicated in yellow and can be found on page 3, line 59 of the revised version of our manuscript.

Reviewer #3 (Remarks to the Author):

The authors have made significant effort to improve the manuscript with the addition of several experiments requested by Reviewers. I have just a few comments:

ANSWER TO REVIEWER: We thank the reviewer for critically reading the revised version of our manuscript and high lightening last issues to be solved.

COMMENT 1: TEXT: The role of hTR capping in promoting the telomerase TMM is well supported. The authors also associate hTR capping with “suppression” of ALT. Forgive me if this is splitting hairs, but to me the word “suppression” implies an active mechanism, which is not necessarily supported by the data. Rather, it may be that – in the absence of hTR capping – telomerase TMM becomes disfavored allowing “characteristics of ALT” or “telomeric substrates conducive to ALT” to become favored. The authors should consider rewording the text to reflect this.

ANSWER TO REVIEWER: We agree with the reviewer. Essentially, instead of using “suppress” we use now the words “antagonize”, “limit” or “restrict”. The changes can be found on:

- Page 2, line 41 (restrict)

- Page 7, line 186 (restrict)
- Page 8, line 205 (antagonize)
- Page 8, line 229 (antagonize)
- Page 8, line 232 (limits)
- Page 11, line 306 (restrict)

COMMENT 2: My personal preference is to refrain from subjective descriptors such as “Importantly”, “Remarkably”, etc.

ANSWER TO REVIEWER: We carried out the suggested modification:

- Importantly: page 4, line 82; changed to: “We found that...”
- Importantly: page 4, line 102; changed to: “Finally...”
- Importantly: page 6, line 157; changed to: “In contrast with data from the G-rich telomere strand, we found that ...”
- Importantly: page 7, line 196; changed to “In this experimental setup...”
- Importantly: page 9, line 241; changed to “In line with these...”
- Importantly: page 9, line 261; word was eliminated from text
- Importantly: page 10, line 286; changed to: “Our study demonstrates that....”
- Remarkably: page 5, line 115; changed to “We found that...”
- Remarkably: page 8, line 208; eliminated from text
- Remarkably: page 8, line 228; eliminated from text
- Remarkably: page 11, line 301; changed to “However, we demonstrate that...”

COMMENT 3:

FIGURES: I appreciate the effort by the authors to pursue further lines of evidence that demonstrate features of ALT. But in Figure 5A, the high abundance of DNA:RNA hybrids (green) make discerning distinct colocalizations with TRF1 unconvincing. And in Figure 5C, the signal/noise is so low as to make this data unconvincing as well.

ANSWER TO REVIEWER: We agree with the reviewer’s comment and present now an improved representative image of DNA:RNA hybrids in experimental cells (**modified Figure 5A**).

In addition, in **modified Figure 5C** we provide now an improved version of the original blot showing more clearly the increase in DNA:RNA hybrids at telomeres in TGS1 loss of functions cells. Both “stripes” shown, originate from the same blot (the entire original blot is shown in the source data file).

COMMENT 4:

In Figure 5L/M (C-circle assay), the data for U2OS should also be plotted in ‘M’ to indicate the relative magnitude of change upon TGS1 knockdown.

ANSWER TO REVIEWER: We have changed the bar diagram in **modified Figure 5M**, that now also shows the quantification of data originating from U2OS cells.

ADDITIONAL MODIFICATION

Change of title of Figure 1: Inhibition of TGS1 blocks 2,2,7-TMG capping in preclinical lung cancer models